# STATISTICAL EFFICIENCY OF SCORE MATCHING: THE VIEW FROM ISOPERIMETRY

**Frederic Koehler**
Stanford University
fkoehler@stanford.edu

**Alexander Heckett**
Carnegie Mellon University
aheckett@andrew.cmu.edu

**Andrej Risteski**
Carnegie Mellon University
aristesk@andrew.cmu.edu

## ABSTRACT

Deep generative models parametrized up to a normalizing constant (e.g. energy-based models) are difficult to train by maximizing the likelihood of the data because the likelihood and/or gradients thereof cannot be explicitly or efficiently written down. Score matching is a training method, whereby instead of fitting the likelihood $\log p(x)$ for the training data, we instead fit the score function $\nabla_x \log p(x)$ — obviating the need to evaluate the partition function. Though this estimator is known to be consistent, its unclear whether (and when) its statistical efficiency is comparable to that of maximum likelihood — which is known to be (asymptotically) optimal. We initiate this line of inquiry in this paper, and show a tight connection between statistical efficiency of score matching and the isoperimetric properties of the distribution being estimated — i.e. the Poincaré, log-Sobolev and isoperimetric constant — quantities which govern the mixing time of Markov processes like Langevin dynamics. Roughly, we show that the score matching estimator is statistically comparable to the maximum likelihood when the distribution has a small isoperimetric constant. Conversely, if the distribution has a large isoperimetric constant — even for simple families of distributions like exponential families with rich enough sufficient statistics — score matching will be substantially less efficient than maximum likelihood. We suitably formalize these results both in the finite sample regime, and in the asymptotic regime. Finally, we identify a direct parallel in the discrete setting, where we connect the statistical properties of pseudolikelihood estimation with approximate tensorization of entropy and the Glauber dynamics.

## 1 INTRODUCTION

Energy-based models (EBMs) are deep generative models parametrized up to a constant of parametrization, namely $p(x) \propto \exp(f(x))$. The primary training challenge is the fact that evaluating the likelihood (and gradients thereof) requires evaluating the partition function of the model, which is generally computationally intractable — even when using relatively sophisticated MCMC techniques. The seminal paper of Song and Ermon (2019) circumvented this difficulty by instead fitting the *score function* of the model, that is $\nabla_x \log p(x)$. Though not obvious how to evaluate this loss from training samples only, Hyvärinen (2005) showed this can be done via integration by parts, and the estimator is consistent (that is, converges to the correct value in the limit of infinite samples).

The maximum likelihood estimator is the de-facto choice for model-fitting for its well-known property of being statistically optimal in the limit where the number of samples goes to infinity (Van der Vaart, 2000). It is unclear how much worse score matching can be — thus, it's unclear how much statistical efficiency we sacrifice for the algorithmic convenience of avoiding partition functions. In the seminal paper (Song and Ermon, 2019), it was conjectured that multimodality, as well as a low-dimensional manifold structure may cause difficulties for score matching. Though the intuition for this is natural: having poor estimates for the score in "low probability" regions of the distribution

can "propagate" into bad estimates for the likelihood once the score vector field is "integrated" — making this formal seems challenging.

We show that the right mathematical tools to formalize, and substantially generalize such intuitions are functional analytic tools that characterize isoperimetric properties of the distribution in question. Namely, we show three quantities, the *Poincaré, log-Sobolev and isoperimetric* constants (which are all in turn very closely related, see Section 2), tightly characterize how much worse the efficiency of score matching is compared to maximum likelihood. These quantities can be (equivalently) viewed as: (1) characterizing the mixing time of *Langevin dynamics* — a stochastic differential equation used to sample from a distribution $p(x) \propto \exp(f(x))$, given access to a gradient oracle for $f$; (2) characterizing "sparse cuts" in the distribution: that is sets $S$, for which the surface area of the set $S$ can be much smaller than the volume of $S$. Notably, multimodal distributions, with well-separated, deep modes have very big log-Sobolev/Poincaré/isoperimetric constants (Gayrard et al., 2004; 2005), as do distributions supported over manifold with negative curvature (Hsu, 2002) (like hyperbolic manifolds). Since it is commonly thought that complex, high dimensional distribution deep generative models are trained to learn do in fact exhibit multimodal and low-dimensional manifold structure, our paper can be interpreted as showing that in many of these settings, score matching may be substantially less statistically efficient than maximum likelihood. Thus, our results can be thought of as a formal justification of the conjectured challenges for score matching in Song and Ermon (2019), as well as a vast generalization of the set of "problem cases" for score matching. This also shows that surprisingly, the same obstructions for efficient inference (i.e. drawing samples from a trained model, which is usual done using Langevin dynamics for EBMs) are also an obstacle for efficient learning using score matching.[1] We roughly show the following results:

1. For *finite number of samples* $n$, we show that if we are trying to estimate a distribution from a class with Rademacher complexity bounded by $\mathcal{R}_n$, as well as a log-Sobolev constant bounded by $C_{LS}$, achieving score matching loss at most $\epsilon$ implies that we have learned a distribution that's no more than $\epsilon C_{LS} \mathcal{R}_n$ away from the data distribution in KL divergence. The main tool for this is showing that the score matching objective is at most a multiplicative factor of $C_{LS}$ away from the KL divergence to the data distribution.

2. In the *asymptotic limit* (i.e. as the number of samples $n \to \infty$), we focus on the special case of estimating the parameters $\theta$ of a probability distribution of an exponential family $\{p_\theta(x) \propto \exp(\langle \theta, F(x) \rangle)$ for some sufficient statistics $F$ using score matching. If the distribution $p_\theta$ we are estimating has Poincaré constant bounded by $C_P$ have asymptotic efficiency that differs by at most a factor of $C_P$. Conversely, we show that if the family of sufficient statistics is sufficiently rich, and the distribution $p_\theta$ we are estimating has isoperimetric constant lower bounded by $C_{IS}$, then the score matching loss is less efficient than the MLE estimator by at least a factor of $C_{IS}$.

3. Based on our new conceptual framework, we identify a precise analogy between score matching in the continuous setting and pseudolikelihood methods in the discrete (and continuous) setting. This connection is made by replacing the Langevin dynamics with its natural analogue — the Glauber dynamics (Gibbs sampler). We show that the *approximation tensorization of entropy inequality* (Marton, 2013; Caputo et al., 2015), which guarantees rapid mixing of the Glauber dynamics, allows us to obtain finite-sample bounds for learning distributions in KL via pseudo-likelihood in an identical way to the log-Sobolev inequality for score matching. A variant of this connection is also made for the related *ratio matching* estimator of Hyvärinen (2007).

4. In Section 7, we perform several simulations which illustrate the close connection between isoperimetry and the performance of score matching. We give examples both when fitting the parameters of an exponential family and when the score function is fit using a neural network.

## 2 Preliminaries

**Definition 1** (Score matching). *Given a smooth ground truth distribution $p$ with sufficient decay at infinity and a smooth distribution $q$, the score matching loss (at the population level) is defined to be*

$$J_p(q) := \frac{1}{2}\mathbb{E}_{X \sim p}[\|\nabla \log p(X) - \nabla \log q(X)\|^2] + K_p = \mathbb{E}_{X \sim p}\left[\text{Tr} \, \nabla^2 \log q + \frac{1}{2}\|\nabla \log q\|^2\right] \quad (1)$$

---

[1]Note, there is another popular variant of score matching called *denoising score matching*, in which the data distribution is convolved with a Gaussian. This will not be the focus of this paper.

*where $K_p$ is a constant independent of q. The last equality is due to Hyvärinen (2005). Given samples from p, the training loss $\hat{J}_p(q)$ is defined by replacing the rightmost expectation with the average over data.*

**Functional and Isoperimetric Inequalities.** Let $q(x)$ be a smooth probability density over $\mathbb{R}^d$. A key role in this work is played by the log-Sobolev, Poincaré, and isoperimetric constants of $q$ — closely related geometric quantities, connected to the mixing of the Langevin dynamics, which have been deeply studied in probability theory and geometric and functional analysis (see e.g. (Gross, 1975; Ledoux, 2000; Bakry et al., 2014)). We discuss the background in more detail in Appendix A.

**Definition 2.** *The* log-Sobolev constant $C_{LS}(q) \geq 0$ *is the smallest constant so that for any smooth probability density p, we have*

$$\mathbf{KL}(p, q) \leq C_{LS}(q)\mathcal{I}(p \mid q) \tag{2}$$

*where* $\mathbf{KL}(p, q) = \mathbb{E}_{X \sim p}[\log(p(X)/q(X))]$ *is the* Kullback-Leibler divergence *or relative entropy and the* relative Fisher information $\mathcal{I}(p \mid q)$ *is defined* [2] *as* $\mathcal{I}(p \mid q) := \mathbb{E}_q \left\langle \nabla \log \frac{p}{q}, \nabla \frac{p}{q} \right\rangle$.

The log-Sobolev inequality is equivalent to exponential ergodicity of the *Langevin dynamics* for $q$, a canonical Markov process which preserves and is used for sampling $q$, described by the Stochastic Differential Equation $dX_t = -\nabla \log q(X_t)\, dt + \sqrt{2}\, dB_t$. The log-Sobolev constant can be bounded for log-concave distributions: if $\mathcal{P}$ is $\alpha$-strongly log concave, then $C_{LS} \leq 1/2\alpha$ by Bakry-Emery theory (Bakry et al., 2014). See Appendix A for details.

For a class of distributions $\mathcal{P}$, we can also define the *restricted log-Sobolev constant* $C_{LS}(q, \mathcal{P})$ to be the smallest constant such that (2) holds under the additional restriction that $p \in \mathcal{P}$ — see e.g. Anari et al. (2021b). For $\mathcal{P}$ an infinitesimal neighborhood of $p$, the restricted log-Sobolev constant of $q$ becomes half of the *Poincaré constant* or inverse spectral gap $C_P(q)$:

**Definition 3.** *The* Poincaré constant $C_P(q) \geq 0$ *is the smallest constant so that that for all smooth functions* $f : \mathbb{R}^d \to \mathbb{R}$,

$$\mathrm{Var}_q(f) \leq C_P(q)\mathbb{E}_q\|\nabla f\|^2. \tag{3}$$

*It is related to the log-Sobolev constant by* $C_P \leq 2C_{LS}$ *(Lemma 3.28 of Van Handel (2014)).*

Both of these inequalities measure the isoperimetric properties of $q$ from the perspective of functions; they are closely related to the *isoperimetric constant*:

**Definition 4.** *The* isoperimetric constant $C_{IS}(q)$ *is the smallest constant, s.t. for every set S,*

$$\min\left\{\int_S q(x)dx, \int_{S^C} q(x)dx\right\} \leq C_{IS}(q) \liminf_{\epsilon \to 0} \frac{\int_{S_\epsilon} q(x)dx - \int_S q(x)dx}{\epsilon}. \tag{4}$$

*where* $S_\epsilon = \{x : d(x, S) \leq \epsilon\}$ *and* $d(x, S)$ *denotes the (Euclidean) distance of x from the set S. The isoperimetric constant is related to the Poincaré constant by* $C_P \leq 4C_{IS}^2$ *(Proposition 8.5.2 of Bakry et al. (2014)). Assuming S is chosen so* $\int_S q(x)dx < 1/2$, *the left hand side can be interpreted as the volume and the right hand side as the surface area of S with respect to q.*

**Mollifiers** We recall the definition of one of the standard mollifiers/bump functions, as used in e.g. Hörmander (2015). Mollifiers are smooth functions useful for approximating non-smooth functions: convolving a function with a mollifier makes it "smoother", in the sense of the existence and size of the derivatives. Precisely, define the (infinitely differentiable) function $\psi : \mathbb{R}^d \to \mathbb{R}$ as $\psi(y) = \frac{1}{G_d}e^{-1/(1-|y|^2)}$ for $|y| < 1$ and $\psi(y) = 0$ for $|y| \geq 1$, where $G_d := \int e^{-1/(1-|y|^2)}dy$. For $\gamma > 0$, we also define a "sharpening" of $\psi$, namely $\psi_\gamma(y) = \gamma^{-d}\psi(y/\gamma)$ so that $\int \psi_\gamma = 1$.

**Notation.** For a random vector $X$, $\Sigma_X := \mathbb{E}[XX^T] - \mathbb{E}[X]\mathbb{E}[X]^T$ denotes its covariance matrix.

## 3 LEARNING DISTRIBUTIONS FROM SCORES: NONASYMPTOTIC THEORY

Though consistency of the score matching estimator was proven in Hyvärinen (2005), it is unclear what one can conclude about the proximity of the learned distribution from a finite number of samples. Precisely, we would like a guarantee that shows that *if the training loss (i.e. empirical estimate*

---

[2]There are several alternatives formulas for $\mathcal{I}(p \mid q)$, see Remark 3.26 of Van Handel (2014).

*of* (1)*) is small, the learned distribution is close to the ground truth distribution* (e.g. in the KL divergence sense). However, this is not always true! We will see an illustrative example where this is not true in Section 7 and also establish a general negative result in Section 4.

Our first new observation is that understanding the multiplicative gap between the KL divergence and the score matching test loss is *equivalent to understanding log-Sobolev constants*. Based on this, we prove (Theorem 1) that minimizing the training loss *does learn the true distribution*, assuming that the class of distributions we are learning have bounded complexity and small log-Sobolev constant. First, we formalize the connection to the log-Sobolev constant:

**Proposition 1.** *The log-Sobolev inequality for $q$ is equivalent to the following inequality over all smooth probability densities $p$:*

$$\mathbf{KL}(p, q) \leq 2C_{LS}(q)(J_p(q) - J_p(p)). \tag{5}$$

*More generally, for a class of distribution $p \in \mathcal{P}$ the restricted log-Sobolev constant is the smallest constant such that $\mathbf{KL}(p, q) \leq C_{LS}(q, \mathcal{P})(J_p(q) - J_p(p))$ for all distributions $p$.*

*Proof.* This follows from the following equivalent form for the relative Fisher information (Shao et al., 2019; Vempala and Wibisono, 2019)

$$\begin{aligned}
\mathcal{I}(p \mid q) &= \mathbb{E}_q \langle \nabla \frac{p}{q}, \nabla \log \frac{p}{q} \rangle \\
&= \mathbb{E}_p \langle \frac{q}{p} \nabla \frac{p}{q}, \nabla \log \frac{p}{q} \rangle = \mathbb{E}_p \langle \nabla \log \frac{p}{q}, \nabla \log \frac{p}{q} \rangle = \mathbb{E}_p \|\nabla \log p - \nabla \log q\|^2. \tag{6}
\end{aligned}$$

Using this and (1) the log-Sobolev inequality can be rewritten as $\mathbf{KL}(p, q) \leq C_{LS}(J_p(q) - J_p(p))$ which proves the first claim, and the same argument shows the second claim. $\square$

**Remark 1** (Interpretation of Score Matching)**.** *The left hand side of (5) is $\mathbf{KL}(p, q) = \mathbb{E}_p[\log p] - \mathbb{E}_p[\log q]$. The first term is independent of $q$ and the second term is the likelihood, the objective for Maximum Likelihood Estimation. So (5) shows that the score matching objective is a* relaxation *(within a multiplicative factor of $C_{LS}(q)$) of maximum-likelihood via the log-Sobolev inequality. We discuss connections to other proposed interpretations in Appendix B.*

**Remark 2.** *Interestingly, the log-Sobolev constant which appears in the bound is that of $q$ and not $p$ the ground truth distribution. This is useful because $q$ is known to the learner whereas $p$ is only indirectly observed. If $q$ is actually close to $p$, the log-Sobolev constants are comparable due to the Holley-Stroock perturbation principle (Proposition 5.1.6 of Bakry et al. (2014)).*

To our knowledge, we are the first to point out the useful connection of the score matching loss with the log-Sobolev inequality. Because log-Sobolev constants have been well-studied, this observation has many nice consequences which would otherwise be difficult to prove. Combining Proposition 1, bounds on log-Sobolev constants from the literature, and generalization theory gives us finite-sample guarantees for learning distributions in KL divergence via score matching. [3]

**Theorem 1.** *Suppose that $\mathcal{P}$ is a class of probability distributions containing $p$ and define $C_{LS}(\mathcal{P}, \mathcal{P}) := \sup_{q \in \mathcal{P}} C_{LS}(q, \mathcal{P}) \leq \sup_{q \in \mathcal{P}} C_{LS}(q)$ to be the worst-case (restricted) log-Sobolev constant in the class of distributions. Let*

$$\mathcal{R}_n := \mathbb{E}_{X_1, \ldots, X_n, \epsilon_1, \ldots, \epsilon_n} \sup_{q \in \mathcal{P}} \frac{1}{n} \sum_{i=1}^n \epsilon_i \left[ \operatorname{Tr} \nabla^2 \log q(X_i) + \frac{1}{2} \|\nabla \log q(X_i)\|^2 \right]$$

*be the expected Rademacher complexity of the class given $n$ samples $X_1, \ldots, X_n \sim p$ i.i.d. and independent $\epsilon_1, \ldots, \epsilon_n \sim Uni\{\pm 1\}$ i.i.d. Rademacher random variables. Let $\hat{p}$ be the score matching estimator from $n$ samples, i.e. $\hat{p} = \arg\min_{q \in \mathcal{P}} \hat{J}_p(q)$. Then*

$$\mathbb{E} \, \mathbf{KL}(p, \hat{p}) \leq 4C_{LS}(\mathcal{P}, \mathcal{P})\mathcal{R}_n.$$

*In particular, if $C_{LS}(\mathcal{P}, \mathcal{P}) < \infty$ then $\lim_{n \to \infty} \mathbb{E} \, \mathbf{KL}(p, \hat{p}) = 0$ as long as $\lim_{n \to \infty} \mathcal{R}_n = 0$.*

---

[3]We use the simplest version of Rademacher complexity bounds to illustrate our techniques. Standard literature, e.g. Shalev-Shwartz and Ben-David (2014); Bartlett et al. (2005) contains more sophisticated versions, and our techniques readily generalize.

*Proof.* By the standard symmetrization argument (Theorem 26.3 of Shalev-Shwartz and Ben-David (2014)) we have $\mathbb{E} J_p(\hat{p}) - J_p(p) \leq 2\mathcal{R}_n$, so by Proposition 1 we have $\mathbb{E} \mathbf{KL}(p, \hat{p}) \leq 2\mathbb{E} C_{LS}(\mathcal{P})(J_p(\hat{p}) - J_p(p)) \leq 4 C_{LS}(\mathcal{P})\mathcal{R}_n.$ □

**Example 1.** *Suppose we are fitting an isotropic Gaussian in $d$ dimensions with unknown mean $\mu^*$ satisfying $\|\mu^*\| \leq R$. The class of distributions $\mathcal{P}$ is $q_\mu$ with $\|\mu\| \leq R$ of the form $q_\mu(x) \propto \exp\left(-\|x - \mu\|^2/2\right)$ so the expected Rademacher complexity can be upper bounded as so:*

$$\mathcal{R}_n = \mathbb{E} \sup_\mu \frac{1}{n} \sum_{i=1}^n \epsilon_i \left[-d/2 + \frac{1}{2}\|X_i - \mu\|^2\right]$$

$$= \mathbb{E} \sup_\mu \left\langle \frac{1}{n} \sum_{i=1}^n \epsilon_i X_i, \mu \right\rangle = R \, \mathbb{E} \left\| \frac{1}{n} \sum_{i=1}^n \epsilon_i X_i \right\| \leq R \sqrt{\mathbb{E} \left\| \frac{1}{n} \sum_{i=1}^n \epsilon_i X_i \right\|^2} = R\sqrt{\frac{R^2 + d}{n}}$$

*where the inequality is Jensen's inequality and in the last step we expanded the square and used that $\mathbb{E}\epsilon_i \epsilon_j = 1(i = j)$ and $\mathbb{E}\|X_i\|^2 \leq R^2 + d$. Recall that the standard Gaussian distribution is 1-strongly log concave so $C_{LS} \leq 1/2$. Hence we have the concrete bound $\mathbb{E} \mathbf{KL}(p, \hat{p}) \leq 2R\sqrt{\frac{R^2+d}{n}}$.*

## 4 STATISTICAL COST OF SCORE MATCHING: ASYMPTOTIC RESULTS

In this section, we compare the asymptotic efficiency of the score matching estimator in exponential families to the effiency of the maximum likelihood estimator. Because we are considering asymptotics, we might expect (recall the discussion in Section 2) that the relevant functional inequality will be the local version of the log-Sobolev inequality around the true distribution $p$, which is the Poincaré inequality for $p$. Our results will show precisely how this occurs and characterize the situations where score matching is substantially less statistically efficient than maximum likelihood.

**Setup.** In this section, we will focus on distributions from exponential families. We will consider estimating the parameters of an exponential family using two estimators, the classical maximum likelihood estimator (MLE), and the score matching estimator; we will use that the score matching estimator $\arg\min_{\theta'} \hat{J}_p(p_{\theta'})$ admits a closed-form formula in this setting.

**Definition 5** (Exponential family). *For sufficient statistics $F: \mathbb{R}^d \to \mathbb{R}^m$, the exponential family of distributions associated with $F$ is $\{p_\theta(x) \propto \exp\left(\langle\theta, F(x)\rangle\right) | \theta \in \Theta \subseteq \mathbb{R}^m\}$.*

**Definition 6** (MLE, Van der Vaart (2000)). *Given i.i.d. samples $x_1, \ldots, x_n \sim p_\theta$, the maximum likelihood estimator is $\hat{\theta}_{MLE} = \arg\max_{\theta' \in \Theta} \hat{\mathbb{E}}\left[\log p_{\theta'}(X)\right]$, where $\hat{\mathbb{E}}$ denotes the expectation over the samples. As $n \to \infty$ and under appropriate regularity conditions, we have $\sqrt{n}\left(\hat{\theta}_{MLE} - \theta\right) \to N(0, \Gamma_{MLE})$, where $\Gamma_{MLE} := \Sigma_F^{-1}$ and $\Sigma_F$ is known as the Fisher information matrix.*

**Proposition 2** (Score matching estimator, Equation (34) of Hyvärinen (2007)). *Given i.i.d. samples $x_1, \ldots, x_n \sim p_\theta$, the score matching estimator equals $\hat{\theta}_{SM} = -\hat{\mathbb{E}}[(JF)_X (JF)_X^T]^{-1}\hat{\mathbb{E}}\Delta F$, where $(JF)_X : m \times d$ is the Jacobian of $F$ at the point $X$, $\Delta f = \sum_i \partial_i^2 f$ is the Laplacian and it is applied coordinate wise to the vector-valued function $F$.*

### 4.1 ASYMPTOTIC NORMALITY

Next, we prove asymptotic normality of the score matching estimator and give a formula for the limiting renormalized covariance matrix $\Gamma_{SM}$.[4] Since the MLE also satisfies asymptotic normality with an explicit covariance matrix, we can then proceed in the next sections to compare their relative efficiency (as in e.g. Section 8.2 of Van der Vaart (2000)) by comparing the asymptotic covariances $\Gamma_{SM}$ and $\Gamma_{MLE}$. The proof of the following result is in Appendix C.

**Proposition 3** (Asymptotic normality). *As $n \to \infty$, and assuming sufficient smoothness and decay conditions so that score matching is consistent (see Hyvärinen (2005)) we have the following convergence in distribution: $\sqrt{n}(\hat{\theta}_{SM} - \theta) \to N(0, \Gamma_{SM})$, where*

$$\Gamma_{SM} := \mathbb{E}[(JF)_X (JF)_X^T]^{-1} \Sigma_{(JF)_X (JF)_X^T \theta + \Delta F} \mathbb{E}[(JF)_X (JF)_X^T]^{-1}. \tag{7}$$

---

[4]Asymptotic normality was proved in Corollary 1 of Song et al. (2020) — we reprove it because in the context of exponential families, we will show and use a much simpler expression for the limiting covariance.

## 4.2 STATISTICAL EFFICIENCY OF SCORE MATCHING UNDER A POINCARÉ INEQUALITY

Our first result will show that if we are estimating a distribution with a small Poincaré constant (and some relatively mild smoothness assumptions), the statistical efficiency of the score matching estimator is not much worse than the maximum likelihood estimator.

**Theorem 2** (Efficiency under a Poincaré inequality). *Suppose the distribution $p_\theta$ satisfies a Poincaré inequality with constant $C_P$. Then we have*

$$\|\Gamma_{SM}\|_{OP} \le 2C_P^2 \|\Gamma_{MLE}\|_{OP}^2 \left( \|\theta\|^2 \mathbb{E}\|(JF)_X\|_{OP}^4 + \mathbb{E}\|\Delta F\|_2^2 \right).$$

*More generally, the same bound holds assuming only the following restricted version of the Poincaré inequality: for any $w$, $\mathrm{Var}(\langle w, F(x)\rangle) \le C_P \mathbb{E}\|\nabla \langle w, F(x)\rangle\|_2^2$.*

**Remark 3.** *To interpret the terms in the bound, the quantities $\mathbb{E}_{p_\theta}\|(JF)_X\|_{OP}^4$ and $\mathbb{E}\|\Delta F\|_2^2$ can be seen as a measure of the smoothness of the sufficient statistics $F$, and $\|\theta\|$ as a bound on the radius of parameters for the exponential family. In Section 7 we will give an example to show bounded smoothness is indeed necessary for score matching to be efficient. A direct consequence of this result (see Appendix D) is that with 99% probability and for sufficiently large $n$, $n\|\theta - \hat\theta_{SM}\|^2 \le \left( n\mathbb{E}\|\theta - \hat\theta_{MLE}\|^2 \right)^2 \cdot O\left( C_P^2 m \left( \|\theta\|^2 \mathbb{E}\|(JF)_X\|_{OP}^4 + \mathbb{E}\|\Delta F\|_2^2 \right) \right)$. So if the the distribution is smooth and Poincaré, score matching achieves small $\ell_2$ error provided MLE does. We illustrate Theorem 2 for a natural example of a bimodal distribution in Example 3 of Appendix G.*

The proof of Theorem 2 is in Appendix D, and the main lemma to prove the theorem is the following:

**Lemma 1.** $\mathbb{E}[(JF)_X(JF)_X^T]^{-1} \preceq C_P \Sigma_F^{-1}$ *where $C_P$ is the Poincaré constant of $p_\theta$.*

*Proof.* For any vector $w \in \mathbb{R}^m$, we have by the Poincaré inequality that

$$C_P \langle w, \mathbb{E}[(JF)_X(JF)_X^T]w\rangle = C_P \mathbb{E}\|\nabla_x \langle w, F(x)\rangle|_X\|_2^2 \ge \mathrm{Var}(\langle w, F(x)\rangle) = \langle w, \Sigma_F w\rangle$$

This shows $C_P \mathbb{E}[(JF)_X(JF)_X^T] \succeq \Sigma_F$ and inverting both sides gives the result. □

## 4.3 STATISTICAL EFFICIENCY LOWER BOUNDS FROM SPARSE CUTS

In this section, we prove a converse to Theorem 2: whereas a small (restricted) Poincaré constant upper bounds the variance of the score matching estimator, if the Poincaré constant of our target distribution is large and we have sufficiently rich sufficient statistics, score matching will be extremely inefficient compared to the MLE. In fact, we will be able to do so by taking an arbitrary family of sufficient statistics, and adding a single sufficient statistic ! Informally, we'll show the following:

*Consider estimating a distribution $p_\theta$ in an exponential family with isoperimetric constant $C_{IS}$. Then, $p_\theta$ can be viewed as a member of an enlarged exponential family with one more ($O_{\partial S}(1)$-Lipschitz) sufficient statistic, such that score matching has asymptotic relative efficiency $\Omega_{\partial S}(C_{IS})$ compared to the MLE, where $\partial S$ denotes the boundary of the isoperimetric cut of $p_\theta$ and $\Omega_{\partial S}$ indicates a constant depending only on the geometry of the manifold $\partial S$.*

As noted in Section 2, a large Poincaré constant implies a large isoperimetric constant — so we focus on showing that the score matching estimator is inefficient when there is a set $S$ which is a "sparse cut". Our proof uses differential geometry, so our final result will depend on standard geometric properties of the boundary $\partial S$ (in terms of how small $\gamma$ is, see Appendix E for more discussion, a proof sketch, as well as the full proof). We now give the formal statement.

**Theorem 3** (Inefficiency of score matching in the presence of sparse cuts). *There exists an absolute constant $c > 0$ so that the following is true. Suppose $S$ is a set with smooth and compact boundary $\partial S$, and suppose that $p_{\theta_1^*}$ is an element of an exponential family with sufficient statistic $F_1$ and parameterized by elements of $\Theta_1$. Define an additional sufficient statistic $F_2 = 1_S * \psi_\gamma$ so that the enlarged exponential family contains distributions with $\theta_1 \in \Theta_1, \theta_2 \in \mathbb{R}$ of the form*

$$p_{(\theta_1, \theta_2)}(x) \propto \exp(\langle \theta_1, F_1(x)\rangle + \theta_2 F_2(x))$$

*and consider the MLE and score matching estimators in this exponential family with ground truth $p_{(\theta_1^*, 0)}$. Suppose that $1_S$ is not an affine function of $F_1$, and so there exists some $\delta_1 > 0$ such that*

$$\sup_{w_1} \mathrm{Cov}\left(\frac{\langle w_1, F_1\rangle}{\sqrt{\mathrm{Var}(\langle w_1, F_1\rangle)}}, \frac{1_S}{\sqrt{\mathrm{Var}(1_S)}}\right)^2 \leq 1 - \delta_1.$$ *Then for all $\gamma$ sufficiently small in terms of $S$, there exists a vector $w$ such that the asymptotic relative (in)efficiency of the score matching estimator compared to the MLE for estimating $\langle w, \theta\rangle$ admits the following lower bound*

$$\frac{\langle w, \Gamma_{SM}w\rangle}{\langle w, \Gamma_{MLE}w\rangle} \geq \frac{c'}{\gamma} \frac{\min\{\Pr(X \in S), \Pr(X \notin S)\}}{\int_{x \in \partial S} p(x)dx}$$

*provided* $c' := c^d \dfrac{1 - \left(\sqrt{1-\delta_1} + 2\sqrt{\frac{\gamma \int_{x \in \partial S} p(x)dx}{\Pr(X \in S)(1 - \Pr(X \in S))}}\right)^2}{1 + \|\Sigma_{F_1}\|_{OP}} > 0.$

**Remark 4.** *If we choose $S$ to be the set achieving the worst isoperimetric constant, then the right hand side of the bound is simply $\frac{c'}{\gamma}C_{IS}$. (See the appendix for details.) Finally, we observe that although $c'$ is exponentially small in $d$, the bound is still useful in high dimensions because in the bad cases of interest $C_{IS}$ is often exponentially large in $d$. For example, this is the case for a mixture of standard Gaussians with $\Omega(\sqrt{d})$ separation between the means (see e.g. Chen et al. (2021a)).*

**Remark 5.** *The assumption $\delta_1 > 0$ is a quantitative way of saying that the function $1_S$, the cut we are using to define the new sufficient statistic $F_2$, is not already a linear combination of the existing sufficient statistics. The assumptions will always holds with some $\delta_1 \geq 0$ by the Cauchy-Schwarz inequality. The equality case is when $1_S$ is an affine function of $\langle w_1, F_1\rangle$ — if such a linear dependence exists, the parameterization is degenerate and the coefficient of $F_2$ is not identifiable.*

**Example 2.** *A concrete example in one dimension with a single sufficient statistic is*

$$F_1(x) = -\frac{1}{8a^2}(x - a)^2(x + a)^2 = -x^4/8a^2 + x^2/4 - a^2/8$$

*and $\theta = (1, 0)$ for a parameter $a > 1$ to be taken large. This looks similar to a mixture of standard Gaussians centered at $-a$ and $a$. Specializing Theorem 3 to this case, we get:*

**Corollary 1.** *There exists absolute constants $\gamma_0 > 0$ and $c > 0$ so that the following is true. Suppose that $a > 1$, $\theta = (1, 0)$, and expanded exponential family $\{p_{\theta'}\}_{\theta'}$ with $p_{\theta'}(x) \propto \exp(\langle \theta', (F_1(x), F_2(x))\rangle)$ and new sufficient statistic $F_2$ is the output of Theorem 3 applied to $F_1$, $S = \{x : x > 0\}$, and $\gamma = \gamma_0$. Then there exists $w$ so that the relative (in)efficiency of estimating $\langle w, \theta\rangle$ is lower bounded as*

$$\frac{\langle w, \Gamma_{SM}w\rangle}{\langle w, \Gamma_{MLE}w\rangle} \geq c\, e^{a^2/8}.$$

# 5 DISCRETE ANALOGUES: PSEUDOLIKELIHOOD, GLAUBER DYNAMICS, AND APPROXIMATE TENSORIZATION

Several authors have proposed variants of score matching for discrete probability distributions, e.g. Lyu (2009); Shao et al. (2019); Hyvärinen (2007). Furthermore, Hyvärinen (2006; 2007) pointed out some connections between pseudolikelihood methods (a classic alternative to maximum likelihood in statistics Besag (1975; 1977)), Glauber dynamics (a.k.a. Gibbs sampler, see Appendix F for the definition), and score matching. Finally, just like the log-Sobolev inequality controls the rapid mixing of Langevin dynamics, there are functional inequalities (Gross, 1975; Bobkov and Tetali, 2006) which bound the mixing time of Glauber dynamics. Thus, we ask: *Is there a discrete analogue of the relationship between score matching and the log-Sobolev inequality?*

The answer is yes. To explain further, we need a key concept recently introduced by Marton (2013; 2015) and Caputo et al. (2015): if $(\Omega_1, \mathcal{F}_1), \ldots (\Omega_d, \mathcal{F}_d)$ are arbitrary measure spaces, we say a distribution $q$ on $\bigotimes_{i=1}^d \Omega_i$ satisfies *approximation tensorization of entropy* with constant $C_{AT}(q)$ if

$$\mathbf{KL}(p, q) \leq C_{AT}(q) \sum_{i=1}^d \mathbb{E}_{X_{\sim i} \sim p_{\sim i}}[\mathbf{KL}(p(X_i \mid X_{\sim i}), q(X_i \mid X_{\sim i}))]. \tag{8}$$

This inequality is sandwiched between two discrete versions of the log-Sobolev inequality (Proposition 1.1 of Caputo et al. (2015)): it is weaker than the standard discrete version of the log-Sobolev inequality (Diaconis and Saloff-Coste, 1996) and stronger than the Modified Log-Sobolev Inequality

(Bobkov and Tetali, 2006) which characterizes exponential ergodicity of the Glauber dynamics. We define a restricted version $C_{AT}(q, \mathcal{P})$ analogously to the restricted log-Sobolev constant.

Finally, we recall the *pseudolikelihood* objective (Besag, 1975) based on entrywise conditional probabilities: $L_p(q) := \sum_{i=1}^{d} \mathbb{E}_{X \sim p}[\log q(X_i \mid X_{\sim i})]$. With these definition in place, we can show that just as matching objective is a relaxation of maximum likelihood through the log-Sobolev inequality, pseudolikelihood is a relaxation through approximate tensorization of entropy:

**Proposition 4.** *We have* $\mathbf{KL}(p, q) \leq C_{AT}(q)(L_p(p) - L_p(q))$ *and more generally for any class* $\mathcal{P}$ *containing* $p$, *we have* $\mathbf{KL}(p, q) \leq C_{AT}(q, \mathcal{P})(L_p(p) - L_p(q))$.

*Proof.* Observe that $L_p(p) - L_p(q) = \sum_{i=1}^{d} \mathbb{E}_{X_{\sim i} \mid p_{\sim i}}[\mathbf{KL}(p(X_i \mid X_{\sim i}), q(X_i \mid X_{\sim i}))]$, so the result follows by expanding the definition. $\square$

**Remark 6.** *Pseudolikelihood methods (and variants like node-wise regression) are one of the dominant approaches to fitting fully-observed graphical models, e.g. (Wu et al., 2019; Lokhov et al., 2018; Klivans and Meka, 2017; Kelner et al., 2020). Like score matching, pseudolikelihood methods do not require computing normalizing constants which can be slow or computationally hard (e.g. Sly and Sun (2012)). Pseudolikelihood is applicable in both discrete and continuous settings, as is our connection with approximate tensorization. An analogous version of Theorem 1 holds by the same argument (Theorem 5 in Appendix F) and guarantees learning in KL when q satisfies approximate tensorization (e.g. under Dobrushin's uniqueness threshold (Marton, 2015)).*

**Remark 7.** *(Hyvärinen, 2007) proposed a version of score matching for distributions on the hypercube $\{\pm 1\}^d$ and observed that the resulting method ("ratio matching") bears similarity to pseudolikelihood. A similar calculation as the proof of Proposition 4 allows us to arrive at ratio matching based on a strengthening of approximate tensorization studied in (Marton, 2015). Our derivation seems more conceptual than the original derivation, explains the similarity to pseudolikelihood, and establishes some useful connections. For space reasons, this is included in Appendix F.2.*

## 6 RELATED WORK

Score matching was originally introduced by Hyvärinen (2005), who also proved that the estimator is asymptotically consistent. In (Hyvärinen, 2007), the authors propose estimators that are defined over bounded domains. (Song and Ermon, 2019) scaled the techniques to neurally parameterized energy-based models, leveraging score matching versions like denoising score matching Vincent (2011), which involves an annealing strategy by convolving the data distribution with Gaussians of different variances, and sliced score matching (Song et al., 2020). The authors conjectured that annealing helps with multimodality and low-dimensional manifold structure in the data distribution — and our paper can be seen as formalizing this conjecture.

The connection between score matching objective and the relative Fisher information in (6) was previously observed in (Shao et al., 2019; Nielsen, 2021). We also remark that since $\mathcal{I}(p|q) = -\frac{d}{dt} \mathbf{KL}(p_t, q) \mid_{t=0}$ for $p_t$ the output of Langevin dynamics at time $t$, score matching can be interpreted as finding a $q$ to minimize the contraction of the Langevin dynamics for $q$ started at $p$. Previously, (Lyu, 2009) observed that the score matching objective can be interpreted as the infinitesimal change in KL divergence as we add Gaussian noise — see Appendix B for an explanation why these two quantities are equal. In the discrete setting, it was recently observed that approximate tensorization has applications to identity testing of distributions in the "coordinate oracle" query model (Blanca et al., 2022), which is another application of approximate tensorization outside of sampling otherwise unrelated to our result. Finally, (Block et al., 2020; Lee et al., 2022) show guarantees on running Langevin dynamics, given estimates on $\nabla \log p$ that are only $\epsilon$-correct in the $L_2(p)$ sense. They show that when the Langevin dynamics are run for some moderate amount of time, the drift between the true Langevin dynamics (using $\nabla \log p$ exactly) and the noisy estimates can be bounded.

## 7 SIMULATIONS

**Exponential family experiments.** *Fitting a bimodal distribution with and without a statistic approximating a cut.* First, we show the result of fitting a bimodal distribution (as in Example 2) from an exponential family. In Figure 1, the difference of the two sufficient statistics we consider

corresponds to the cut statistic used in our negative result (Theorem 3). As predicted (Corollary 1) score matching performs poorly compared to the MLE as the distance between modes grows.

In Appendix G, we show that when the second sufficient statistic (which is correlated with a sparse cut in the distribution) is removed, score matching performs nearly as well as MLE. This is what our theory predicts (since the cut statistic is removed) and illustrates the use of restricted functional inequalities (in Theorem 2, the restricted Poincaré inequality explains what happens here — see the appendix).

*Fitting a unimodal but not smooth distribution.* For space reasons this is left to Appendix G — we demonstrate that, even if the distribution is unimodal, the performance of score matching degrades as the sufficient statistics become less smooth. Hence the dependence on smoothness in our results, e.g. Theorem 2, is really required.

**Fitting a mixture of Gaussians with a one-layer network.** We also show that empirically, our results are robust even beyond exponential families. In Figure 2 we show the results of fitting a mixture of two Gaussians via score matching[5], where the score function is param-

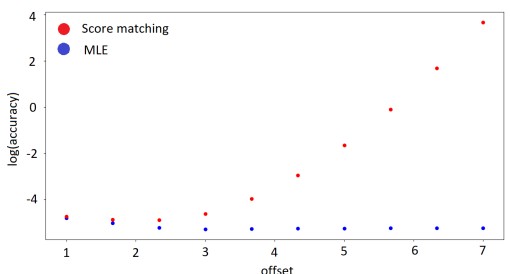

Figure 1: Statistical efficiency of score matching vs MLE for fitting the distribution with ground truth parameters $(\theta_0, \theta_1) = (1, 0)$ of the form $p_\theta(x) \propto e^{\theta_0(x^2 - x^4/(2a^2)) + \theta_1(x^2 - x^4/(2a^2) + \mathrm{erf}(x))}$ as we vary the offset $a$ between 1 and 7 and train with fixed number of samples ($10^6$). We see score matching (red) performs very poorly compared to the MLE (blue) as the offset (distance between modes) grows, by plotting the log of the Euclidean distance to the true parameter for both estimators.

eterized as a one hidden-layer network with tanh activations. We see that the predictions of our theory persist: the distribution is learned successfully when the two modes are close and is not when the modes are far. This matches our expectations, since the Poincaré, log-Sobolev, and isoperimetric constants blow up exponentially in the distance between the two modes (see e.g. Chen et al. (2021a)) and the neural network is capable of detecting the cut between the two modes. We discuss the interpretation of this result more in the appendix.

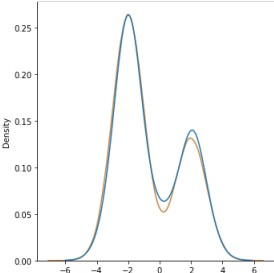
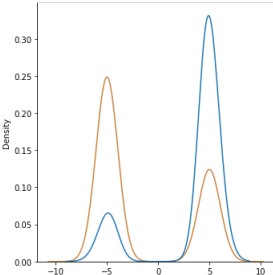

Figure 2: Training a single hidden-layer network to score match a mixture of Gaussians (ground truth green, score matching output blue) succeeds at learning the distribution when the modes are close (left, small isoperimetric constant), but not when they are distant (right, large isoperimetric constant) in which case it weighs the modes incorrectly.

## 8    CONCLUSION

In this paper, we initiate the study of the statistical efficiency of score matching, and identified a close connection to functional inequalities which characterize the ergodicity of Langevin dynamics. For future work, it would be interesting to characterize formally the improvements conferred by annealing strategies like (Song and Ermon, 2019), like it has been done in the setting of sampling using Langevin dynamics (Lee et al., 2018).

---

[5]We note that this experiment is similar in flavor to plots in (Figure 2) in Song and Ermon (2019), where they show that the score is estimated poorly near the low-probability regions of a mixture of Gaussians. In our plots, we numerically integrate the estimates of the score to produce the pdf of the estimated distribution.

**Acknowledgements:** Frederic Koehler acknowledges support by NSF award CCF-1704417, NSF award IIS-1908774, and N. Anari's Sloan Research Fellowship. Andrej Risteski acknowledges support by NSF awards IIS-2211907 and CCF-2238523, an Amazon Research Award on "Causal + Deep Out-of-Distribution Learning".

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

## A    FURTHER BACKGROUND

**Correspondence between functional inequalities and exponential ergodicity.** If $p_t$ is the distribution of the continuous-time Langevin Dynamics[6] for $q$ started from $X_0 \sim p$, then $\mathcal{I}(p \mid q) = -\frac{d}{dt} \mathbf{KL}(p_t, q) \mid_{t=0}$ and so by integrating $\mathbf{KL}(p_t, q) \leq e^{-t/C_{LS}} \mathbf{KL}(p, q)$. This holding for any $p$ and $t$ is an equivalent characterization of the log-Sobolev constant (Theorem 3.20 of Van Handel (2014)). Similarly, the Poincaré inequality implies exponential ergodicity for the $\chi^2$-divergence $\chi^2(p_t, q) \leq e^{-2t/C_P} \chi^2(p, q)$, and this holding for every $p$ and $t$ is an equivalent characterization of the Poincaré constant (Theorem 2.18 of Van Handel (2014)).

We can equivalently view the Langevin dynamics in a functional-analytic way through its definition as a Markov semigroup, which is equivalent to the SDE definition via the Fokker-Planck equation (Van Handel, 2014; Bakry et al., 2014). From this perspective, we can write $p_t = qH_t\frac{p}{q}$ where $H_t$ is the Langevin semigroup for $q$, so $H_t = e^{tL}$ with generator

$$Lf = \langle \nabla \log q, \nabla f \rangle + \Delta f.$$

In this case, the Poincaré constant has a direct interpretation in terms of the inverse spectral gap of $L$, i.e. the inverse of the gap between its two largest eigenvalues.

**Further remarks.** A strengthened isoperimetric inequality (Bobkov inequality) upper bounds the log-Sobolev constant, see Ledoux (2000); Bobkov (1997).

---

[6]See e.g. Vempala and Wibisono (2019) for more background and the connection to the discrete time dynamics.

**Facts about the mollifier $\psi$.** We will use the basic estimate $8^{-d}B_d < G_d < B_d$ where $B_d$ is the volume of the unit ball in $\mathbb{R}^d$, which follows from the fact that $e^{-1/(1-|y|^2)} \geq 1/4$ for $\|y\| \leq 1/2$ and $e^{-1/(1-|y|^2)} \leq 1$ everywhere. It is infinitely differentiable and its gradient is

$$\nabla_y \psi(y) = -(2/G_d)e^{-1/(1-\|y\|^2)}\frac{y}{(1-\|y\|^2)^2} = \frac{-2y}{(1-\|y\|^2)^2}\psi(y)$$

It is straightforward to check that $\sup_y \|\nabla_y \psi(y)\| < 1/G_d$. For $\gamma > 0$, we'll also define a "sharpening" of $\psi$, namely $\psi_\gamma(y) = \gamma^{-d}\psi(y/\gamma)$ so that $\int \psi_\gamma = 1$ and (by chain rule)

$$\nabla_y \psi_\gamma(y) = \gamma^{-d-1}(\nabla\psi)(y/\gamma) = \frac{-2y/\gamma^2}{(1-\|y/\gamma\|^2)^2}\psi_\gamma(y/\gamma)$$

so in particular $\|\nabla_y \psi_\gamma\|_2 \leq \gamma^{-d-1}/G_d$.

**Reach and Condition Number of a Manifold.** For a smooth submanifold $\mathcal{M}$ of Euclidean space, the *reach* $\tau_\mathcal{M}$ is the smallest radius $r$ so that every point with distance at most $r$ to the manifold $\mathcal{M}$ has a unique nearest point on $\mathcal{M}$ (Federer, 1959); the reach is guaranteed to be positive for compact manifolds. The reach has a few equivalent characterizations (see e.g. Niyogi et al. (2008)); a common terminology is that the *condition number* of a manifold is $1/\tau_\mathcal{M}$.

## B  RECOVERING LYU'S INTERPRETATION OF SCORE MATCHING

As mentioned, the connection between score matching objective and the relative Fisher information was previously observed, for example in (Shao et al., 2019; Nielsen, 2021). We also remark that if we use the fact $\mathcal{I}(p|q) = -\frac{d}{dt}\mathbf{KL}(p_t, q)\mid_{t=0}$, the score matching objective has a natural interpretation in terms of select $q$ to minimize the contraction of the Langevin dynamics for $q$ started at $p$. On the other hand, Lyu (2009) previously observed that the score matching objective can be interpreted as the infinitesimal change in KL divergence between $p$ and $q$ as we add noise to both of them. We now explain why these two quantities are equal by giving a proof of their equality (which is shorter than the one you get by going through the proof in Lyu (2009)).

Before giving the formal proof, we give some intuition for why the statement should be true. The Langevin dynamics approximately adds a noise of size $N(0, 2t)$ and subtracts a gradient step along $\nabla \log q$, and this dynamics preserves $q$. For small $t$, the gradient step is essentially reversible and preserves the KL. So heuristically, reversing the gradient step gives $KL(p_t, q) \approx KL(N(0, 2t) * p, N(0, 2t) * q)$. We now give the formal proof.

**Lemma 2.** *Assuming smooth probability densities $p(x)$ and $q(x)$ decay sufficiently fast at infinity,*

$$\frac{d}{dt}KL(p_t, q)\Big|_{t=0} = \frac{d}{dt}KL(p * N(0, 2t), q * N(0, 2t))\Big|_{t=0}$$

*where $*$ denotes convolution.*

*Proof.* Recalling from Appendix A that $H_t = e^{tL}$ we have that $\frac{d}{dt}\frac{p_t}{q} = \frac{d}{dt}H_t\frac{p}{q} = L\frac{p}{q}$. Since $KL(p_t, q) = \mathbb{E}_q[\frac{p_t}{q}\log\frac{p_t}{q}]$ and $\frac{d}{dx}[x \log x] = \log x + 1$, it follows by the chain rule that

$$\frac{d}{dt}KL(p_t, q) = \mathbb{E}_q\left[\left(\log\frac{p}{q}+1\right)L\frac{p}{q}\right] = \mathbb{E}_q\left[\left(\log\frac{p}{q}+1\right)\left(\langle\nabla\log q, \nabla\frac{p}{q}\rangle + \Delta\frac{p}{q}\right)\right]$$

$$= \mathbb{E}_q\left[\left(\log\frac{p}{q}+1\right)\left(-\langle\nabla\log q, \nabla\frac{p}{q}\rangle + \frac{\Delta p}{q} - \frac{p\Delta q}{q^2}\right)\right]$$

where in the last step we used the quotient rule $\Delta\frac{p}{q} = \frac{\Delta p}{q} - 2\left\langle\nabla\log q, \nabla\frac{p}{q}\right\rangle - \frac{p\Delta q}{q^2}$. On the other hand, by using the Fokker-Planck equation $\frac{\partial}{\partial t}(p * N(0, 2t)) = \Delta p$ (Lemma 2 of Lyu (2009)) and the chain rule we have

$$\frac{d}{dt}KL(p * N(0, 2t), q * N(0, 2t)) = \frac{d}{dt}\int(q * N(0, 2t))\frac{p * N(0, 2t)}{q * N(0, 2t)}\log\frac{p * N(0, 2t)}{q * N(0, 2t)}dx$$

$$= \int(\Delta q)\frac{p}{q}\log\frac{p}{q}dx + \mathbb{E}_q\left[\left(\log\frac{p}{q}+1\right)\left(\frac{\Delta p}{q} - \frac{p\Delta q}{q^2}\right)\right]$$

Since by the chain rule and integration by parts we have

$$\mathbb{E}_q\left[\left(\log\frac{p}{q}+1\right)\left\langle\nabla\log q,\nabla\frac{p}{q}\right\rangle\right] = \int\left[\left\langle\nabla q,\nabla\frac{p}{q}\log\frac{p}{q}\right\rangle\right]dx = -\int(\Delta q)\frac{p}{q}\log\frac{p}{q}dx,$$

we see that the two derivatives are indeed equal. □

## C  PROOF OF PROPOSITION 3

*Proof.* From Hyvärinen (2005), we have consistency of score matching (Theorem 2) and in particular the formula

$$\theta = -\mathbb{E}[(JF)_X(JF)_X^T]^{-1}\mathbb{E}\Delta F. \tag{9}$$

We now compute the limiting distribution of the estimator as the number of samples $n \to \infty$. We will need to use some standard results from probability theory such as Slutsky's theorem and the central limit theorem, see e.g. Van der Vaart (2000) or Durrett (2019) for references. To minimize ambiguity, let $\hat{\mathbb{E}}_n$ denote the empirical expectation over $n$ i.i.d. samples samples and let $\hat{\theta}_n$ denote the score matching estimator $\hat{\theta}_{\text{SM}}$ from $n$ samples. Define $\delta_{n,1}$ and $\delta_{n,2}$ by the equations

$$\hat{\mathbb{E}}_n[(JF)_X(JF)_X^T] = \mathbb{E}[(JF)_X(JF)_X^T] + \delta_{n,1}/\sqrt{n}$$

and

$$\hat{\mathbb{E}}_n\Delta F = \mathbb{E}\Delta f + \delta_{n,2}/\sqrt{n}.$$

By the central limit theorem, $\delta_n = (\delta_{n,1},\delta_{n,2})$ converges in distribution to a multivariate Gaussian (with a covariance matrix that we won't need explicitly) as $n \to \infty$. From the definition

$$\hat{\theta}_n = -\hat{\mathbb{E}}_n[(JF)_X(JF)_X^T]^{-1}\hat{\mathbb{E}}\Delta F$$
$$= -[\mathbb{E}[(JF)_X(JF)_X^T]^{-1}\hat{\mathbb{E}}_n[(JF)_X(JF)_X^T]]^{-1}\mathbb{E}[(JF)_X(JF)_X^T]^{-1}\hat{\mathbb{E}}\Delta F$$

and we now simplify the expression on the right hand side. By applying (9) we have

$$\mathbb{E}[(JF)_X(JF)_X^T]^{-1}\hat{\mathbb{E}}_n\Delta F = \mathbb{E}[(JF)_X(JF)_X^T]^{-1}(\mathbb{E}\Delta F + \delta_{n,2}/\sqrt{n})$$
$$= -\theta + \mathbb{E}[(JF)_X(JF)_X^T]^{-1}\delta_{n,2}/\sqrt{n}$$

Since

$$\mathbb{E}[(JF)_X(JF)_X^T]^{-1}\hat{\mathbb{E}}_n[(JF)_X(JF)_X^T] = I + \mathbb{E}[(JF)_X(JF)_X^T]^{-1}\delta_{n,1}/\sqrt{n}$$

and $(I+X)^{-1} = I - X + X^2 - \cdots$ we have by applying Slutsky's theorem that

$$\mathbb{E}[(JF)_X(JF)_X^T]^{-1}\hat{\mathbb{E}}_n[(JF)_X(JF)_X^T]]^{-1} = I - \mathbb{E}[(JF)_X(JF)_X^T]^{-1}\delta_{n,1}/\sqrt{n} + O_P(1/n)$$

where we use the standard notation $Y_n = O_P(1/n)$ to indicate that $nY_n/f(n) \to 0$ in probability for any function $f$ with $f(n) \to \infty$. Hence

$$\hat{\theta}_{\text{SM}} = -[\mathbb{E}[(JF)_X(JF)_X^T]^{-1}\hat{\mathbb{E}}_n[(JF)_X(JF)_X^T]]^{-1}\mathbb{E}[(JF)_X(JF)_X^T]^{-1}\hat{\mathbb{E}}_n\Delta F$$
$$= -\left[I - \mathbb{E}[(JF)_X(JF)_X^T]^{-1}\delta_{n,1}/\sqrt{n} + O_P(1/n)\right](-\theta + \mathbb{E}[(JF)_X(JF)_X^T]^{-1}\delta_{n,2}/\sqrt{n})$$

and applying Slutsky's theorem again, we find

$$\sqrt{n}(\hat{\theta}_n - \theta) = \mathbb{E}[(JF)_X(JF)_X^T]^{-1}(-\delta_{n,1}\theta - \delta_{n,2}) + O_P(1/\sqrt{n})$$

From the definition, we know

$$\frac{1}{\sqrt{n}}(\delta_{n,1}\theta - \delta_{n,2}) = \hat{\mathbb{E}}_n[-(JF)_X(JF)_X^T\theta - \Delta F] - \mathbb{E}[-(JF)_X(JF)_X^T\theta - \Delta F]$$

so altogether by the central limit theorem, we have

$$\sqrt{n}(\hat{\theta}-\theta) \to N\left(0, \mathbb{E}[(JF)_X(JF)_X^T]^{-1}\Sigma_{(JF)_X(JF)_X^T\theta+\Delta F}\mathbb{E}[(JF)_X(JF)_X^T]^{-1}\right)$$

as claimed.

□

## D    PROOF OF THEOREM 2

First, we will need the following helper lemma:

**Lemma 3.** *For any random vectors $A, B$ we have $\Sigma_{A+B} \preceq 2\Sigma_A + 2\Sigma_B$.*

*Proof.* For any vector $w$ we have

$$\begin{aligned}
\mathrm{Var}(\langle w, A + B\rangle) &= \mathrm{Var}(\langle w, A\rangle) + 2\mathrm{Cov}(\langle w, A\rangle\langle w, B\rangle) + \mathrm{Var}(\langle w, B\rangle) \\
&\leq \mathrm{Var}(\langle w, A\rangle) + 2\sqrt{\mathrm{Var}(\langle w, A\rangle)\mathrm{Var}(\langle w, B\rangle)} + \mathrm{Var}(\langle w, B\rangle) \\
&\leq 2\mathrm{Var}(\langle w, A\rangle) + 2\mathrm{Var}(\langle w, B\rangle)
\end{aligned}$$

where the first inequality is Cauchy-Schwarz for variance and the second is $ab \leq a^2/2 + b^2/2$. We proved for this for every vector which proves the PSD inequality. $\qquad\square$

With this in mind, we can proceed to the proof of Theorem 2:

*Proof of Theorem 2.* Recall from Proposition 3 that

$$\Gamma_{\mathrm{SM}} := \mathbb{E}[(JF)_X(JF)_X^T]^{-1}\Sigma_{(JF)_X(JF)_X^T\theta+\Delta F}\mathbb{E}[(JF)_X(JF)_X^T]^{-1}.$$

By Lemma 1 and submultiplicativity of the operator norm, we have

$$\begin{aligned}
&\|\mathbb{E}[(JF)_X(JF)_X^T]^{-1}\Sigma_{(JF)_X(JF)_X^T\theta+\Delta F}\mathbb{E}[(JF)_X(JF)_X^T]^{-1}\|_{OP} \\
&\leq C_P^2\|\Sigma_F^{-1}\|_{OP}^2\|\Sigma_{(JF)_X(JF)_X^T\theta+\Delta F}\|_{OP}.
\end{aligned}$$

We will finally bound the two operator norms on the right hand side. By Lemma 3, we have

$$\Sigma_{(JF)_X(JF)_X^T\theta+\Delta F} \preceq 2\Sigma_{(JF)_X(JF)_X^T\theta} + 2\Sigma_{\Delta F}$$

Furthermore, we have

$$\|\Sigma_{(JF)_X(JF)_X^T\theta}\|_{OP} \leq \|\mathbb{E}[(JF)_X(JF)_X^T\theta\theta^T(JF)_X(JF)_X^T]\|_{OP} \leq \mathbb{E}\|(JF)_X\|_{OP}^4\|\theta\|^2$$

and

$$\|\Sigma_{\Delta F}\|_{OP} \leq \|\mathbb{E}(\Delta F)(\Delta F)^T\|_{OP} \leq \mathrm{Tr}\,\mathbb{E}(\Delta F)(\Delta F)^T \leq \mathbb{E}\|\Delta F\|_2^2$$

which implies the statement of the theorem. $\qquad\square$

**Supporting details for remark after Theorem 2.**    Since $\sqrt{n}(\theta - \hat{\theta}_{\mathrm{SM}}) \to N(0, \Gamma_{\mathrm{SM}})$ by Proposition 3, for all sufficiently large $n$ it follows from Markov's inequality that with probability at least 99%,

$$n\|\theta - \hat{\theta}_{\mathrm{SM}}\|^2 = O(\mathbb{E}_{Z\sim N(0,\Gamma_{\mathrm{SM}})}\|Z\|^2) = O(\mathrm{Tr}\,\Gamma_{\mathrm{SM}}) = O(m\|\Gamma_{\mathrm{SM}}\|_{OP}).$$

On the other hand, by Fatou's lemma we have that

$$\liminf_{n\to\infty} n\mathbb{E}\|\theta - \hat{\theta}_{\mathrm{MLE}}\|^2 \geq \mathbb{E}_{Z\sim N(0,\Gamma_{\mathrm{MLE}})}\|Z\|^2 = \mathrm{Tr}(\Gamma_{\mathrm{MLE}}) \geq \|\Gamma_{\mathrm{MLE}}\|_{OP}$$

where in the first expression $\hat{\theta}_{\mathrm{MLE}}$ implicitly depends on $n$, the number of samples. Combining these two observations with Theorem 2 and gives the inequality stated in the remark.

## E    PROOF OF THEOREM 3 AND APPLICATIONS

We restate Theorem 3 for the reader's convenience and in a slightly more explicit form in terms of the bounds on $\gamma$. Note that we use the concept of the reach $\tau_{\mathcal{M}}$ of a manifold which was defined in the preliminaries (Appendix A).

**Theorem 4** (Inefficiency of score matching in the presence of sparse cuts, Restatement of Theorem 3). *There exists an absolute constant $c > 0$ such that the following is true. Suppose that $p_{\theta_1^*}$ is an element of an exponential family with sufficient statistic $F_1$ and parameterized by elements of $\Theta_1$. Suppose $S$ is a set with smooth and compact boundary $\partial S$. Let $\tau_{\partial S} > 0$ denote the reach of $\partial S$ (see Appendix A) Suppose that $1_S$ is not an affine function of $F_1$, so there exists $\delta_1 > 0$ such that*

$$\sup_{w_1: \text{Var}(\langle w_1, F_1 \rangle) = 1} \text{Cov}\left(\langle w_1, F_1 \rangle, \frac{1_S}{\sqrt{\text{Var}(1_S)}}\right)^2 \leq 1 - \delta_1. \tag{10}$$

*Suppose that $\gamma > 0$ satisfies $\gamma < \min\left\{\frac{c^d}{(1+\|\theta_1\|)\sup_{x:d(x,\partial S)\leq\gamma}\|(JF_1)_x\|_{OP}}, c\frac{\tau_{\partial S}}{d}\right\}$ and is small enough so that $0 < \delta := 1 - \left(\sqrt{1-\delta_1} + 2\sqrt{\frac{\gamma\int_{x\in\partial S}p(x)dx}{\Pr(X\in S)(1-\Pr(X\in S))}}\right)^2$. Define an additional sufficient statistic $F_2 = 1_S * \psi_\gamma$ so that the enlarged exponential family contains distributions of the form*

$$p_{(\theta_1,\theta_2)}(x) \propto \exp(\langle\theta_1, F_1(x)\rangle + \theta_2 F_2(x))$$

*and consider the MLE and score matching estimators in this exponential family with ground truth $p_{(\theta_1^*, 0)}$.*

*Then the asymptotic renormalized covariance matrix $\Gamma_{MLE}$ of the MLE is bounded above as $\Gamma_{MLE} \preceq \frac{1}{1-\delta}\begin{bmatrix} \Sigma_{F_1}^{-1} & 0 \\ 0 & \frac{1}{\Pr(X\in S)(1-\Pr(X\in S))} \end{bmatrix}$ and there there exists some $w$ and corresponding asymptotic variances $\sigma_{SM}^2(w), \sigma_{MLE}^2(w)$ so that*

$$\sqrt{n}\langle w, \hat{\theta}_{SM} - \theta\rangle \rightarrow N(0, \sigma_{SM}^2(w)), \qquad \sqrt{n}\langle w, \hat{\theta}_{MLE} - \theta\rangle \rightarrow N(0, \sigma_{MLE}^2(w))$$

*and the relative (in)efficiency of the score matching estimator compared to the MLE for estimating $\langle w, \theta\rangle$ admits the following lower bound*

$$\frac{\sigma_{SM}^2(w)}{\sigma_{MLE}^2(w)} \geq \frac{c'}{\gamma}\frac{\min\{\Pr(X\in S), \Pr(X\notin S)\}}{\int_{x\in\partial S}p(x)dx}$$

*where $c' := \frac{\delta c^d}{1+\|\Sigma_{F_1}\|_{OP}}$.*

The proof will proceed in two parts: we will lower bound $\sigma_{SM}^2(w)$ and upper bound $\sigma_{MLE}^2(w)$. The former part will proceed by proving a lower bound on the spectral norm of $\Gamma_{SM}$ (Subsection E.1) — by picking a direction in which the quadratic form is large. The upper bound on $\sigma_{MLE}^2(w)$ (Subsection E.2) will proceed by relating the Fisher matrix for the augmented sufficient statistic $(F_1, F_2)$ with the Fisher matrix for the original sufficient statistic $F_1$.

**Supporting details for remarks after Theorem 3.** If we choose $S$ to be the worst set in the isoperimetric inequaltiy, the term $\frac{\min\{\Pr(X\in S), \Pr(X\notin S)\}}{\int_{x\in\partial S}p(x)dx}$ in the bound is simply $C_{IS}$. To see this, observe that $\lim_{\epsilon\to 0}\frac{\int_{S_\epsilon}p(x)dx - \int_S p(x)dx}{\epsilon} = \int_{x\in\partial S}p(x)dx$ as a special case of Weyl's tube formula (Weyl, 1939; Gray, 2003).

### E.1 LOWER BOUNDING THE SPECTRAL NORM OF $\Gamma_{SM}$

We recall the new statistic $F_2$, defined in terms of the mollifier $\psi$ introduced in Section 2:

$$F_2(x) := (1_S * \psi_\gamma)(x) = \int_{\mathbb{R}^d} 1_S(y)\psi_\gamma(x-y)dy = \int_S \psi_\gamma(x-y)dy$$

and the new sufficient statistic is $F(x) = (F_1(x), F_2(x))$. We first show the following lower bound on the largest eigenvalue of $\Gamma_{SM}$, the renormalized limiting covariance of score matching:

**Lemma 4** (Largest eigenvalue of $\Gamma_{SM}$). *The largest eigenvalue of $K$ satisfies*

$$\lambda_{max}(\Gamma_{SM}) \geq \frac{8^{-d}\gamma^2}{\Pr[d(X,\partial S)\leq\gamma]}\frac{\mathbb{E}_{X|d(X,\partial S)\leq\gamma}\left((\nabla F_2)_X^T(JF)_X^T\theta + \Delta F_2\right)^2}{\sup_{d(x,\partial S)\leq\gamma}\|(JF)_x\|_{OP}^2}. \tag{11}$$

*Proof.* We have

$$\nabla_x F_2(x) = \int_S \nabla_x \psi_\gamma(x-y)dy, \qquad \nabla_x^2 F_2(x) = \int_S \nabla_x^2 \psi_\gamma(x-y)dy.$$

Defining

$$u := \mathbb{E}[(JF)_X(JF)_X^T](0,1) = \mathbb{E}[(JF)_X \nabla_x F_2(x)]$$

we have, by the variational characterization of eigenvalues of symmetric matrices, that

$$\lambda_{max}(K) \geq \frac{\langle u, \mathbb{E}[(JF)_X(JF)_X^T]^{-1} \Sigma_{(JF)_X(JF)_X^T\theta+\Delta F} \mathbb{E}[(JF)_X(JF)_X^T]^{-1} u \rangle}{\|u\|_2^2}. \tag{12}$$

To upper bound the denominator we observe that if $B_d$ is the volume of the unit ball,

$$\|\nabla_x F_2(x)\|_2 = \left\| \int_S (\nabla \psi_\gamma)(x-y)dy \right\|_2 \tag{13}$$

$$\leq 1(d(x,\partial S) \leq \gamma)\gamma^{-d-1} vol(B(X,\gamma))/G_d \tag{14}$$

$$\leq 8^d 1(d(x,\partial S) \leq \gamma)\gamma^{-1} \tag{15}$$

and so

$$\|u\|_2 \leq 8^d \gamma^{-1} \Pr[d(X,\partial S) \leq \gamma] \sup_{d(x,\partial S) \in [-\gamma,\gamma]} \|(JF)_x\|_{OP}$$

where we used the computation of the derivative of $\psi_\gamma$. To lower bound the numerator we have

$$\langle u, \mathbb{E}[(JF)_X(JF)_X^T]^{-1} \Sigma_{(JF)_X(JF)_X^T\theta+\Delta F} \mathbb{E}[(JF)_X(JF)_X^T]^{-1} u \rangle$$

$$= (0,1)^T \Sigma_{(JF)_X(JF)_X^T\theta+\Delta F}(0,1)$$

$$= \mathbb{E}\langle (0,1),(JF)_X(JF)_X^T\theta+\Delta F \rangle^2 = \mathbb{E}\left( (\nabla_x F_2)_X^T(JF)_X^T\theta+\Delta F_2 \right)^2.$$

The integrand is zero except when $d(X,\partial S) \leq \gamma$ so it equals

$$\Pr[d(X,\partial S) \leq \gamma]\mathbb{E}_{X|d(X,\partial S) \in [-\gamma,\gamma]}\left( (\nabla F_2)_X^T(JF)_X^T\theta+\Delta F_2 \right)^2$$

and combining gives the result. $\qquad\square$

We now estimate the right hand side of (11) for small $\gamma$, using differential geometric techniques. The main idea is that as we take $\gamma$ smaller, we end up zooming into the manifold $\partial S$ which locally looks closer and closer to being flat. Differential-geometric quantities describing the manifold appear when we make this approximation rigorous. The most involved term to handle ends up to be calculating the expectation $\mathbb{E}_{X|d(X,\partial S) \leq \gamma}\left( (\nabla F_2)_X^T(JF)_X^T\theta+\Delta F_2 \right)^2$. To do this, we first argue that the term with the Laplacian dominates as $\gamma \to 0$, then by Stokes theorem, we end up integrating $\langle \nabla\psi, dN \rangle$ over intersections of $S$ with small spheres of radius $\gamma$, where $N$ is a normal to $S$. Such quantities can be calculated by comparing to the "flat" manifold case — i.e. when $N$ does not change. How far away these quantities are (thus how small $\gamma$ needs to be) depends on the curvature of $S$ (or more precisely, the condition number of the manifold). Lemma 6 makes rigorous the statement that well-conditioned manifolds are locally flat and then Lemma 7, which is part of the proof of Weyl's tube formula (Gray, 2003; Weyl, 1939), lets us rigorously say that the tubular neighborhood (that is, a thickening of the manifold) behaves similarly to the flat case.

**Lemma 5.** *There exists an absolute constant $c > 0$ such that the following is true. For any $\gamma > 0$ satisfying*

$$\gamma < \min\left\{ \frac{c^d}{(1+\|\theta_1\|)\sup_{x:d(x,\partial S)\leq\gamma}\|(JF_1)_x\|_{OP}}, c\frac{\tau_{\partial S}}{d} \right\}$$

*for score matching on the extended family with $m+1$ sufficient statistics and distribution $p_\theta$ with $\theta = (\theta_1,0)$ we have*

$$\lambda_{max}(\Gamma_{SM}) \geq \frac{c^d}{\gamma \int_{\partial S} p(x)dA}$$

*Proof.* In the denominator, we can observe by (15) that

$$\|(JF)_x\|_{OP}^2 \le \|JF_1\|_{OP}^2 + \|\nabla F_2\|_2^2 \le \|JF_1\|_{OP}^2 + \gamma^{-2}B_d^2 \le 2\gamma^{-2}B_d^2$$

where the last inequality holds assuming $\gamma$ is sufficiently small that $\|JF_1\|_{OP}^2 \le \gamma^{-2}B_d^2$.

In the numerator we can observe

$$(\nabla_x F_2)_X^T (JF)_X^T \theta + \Delta F_2$$
$$= \int_S \langle (\nabla \psi_\gamma)((X-y)), (JF)_X^T \theta \rangle + (\Delta \psi_\gamma)(X-y) dy$$
$$= \int_{S \cap B(X, \gamma)} \langle (\nabla \psi_\gamma)((X-y)), (JF)_X^T \theta \rangle + (\Delta \psi_\gamma)(X-y) dy$$
$$= \gamma^d \int_{B(0,1) \cap (X-S)/\gamma} \langle (\nabla \psi_\gamma)(\gamma u), (JF)_X^T \theta \rangle + (\Delta \psi_\gamma)(\gamma u) du$$
$$= \int_{B(0,1) \cap (X-S)/\gamma} \gamma^{-1} \langle \nabla \psi(u), (JF)_X^T \theta \rangle + \gamma^{-2}(\Delta \psi)(u) du$$
$$= \int_{B(0,1) \cap (X-S)/\gamma} \gamma^{-1} \langle \nabla \psi(u), (JF)_X^T \theta \rangle + \int_{\partial(B(0,1) \cap (X-S)/\gamma)} \gamma^{-2} \langle \nabla \psi, dN \rangle$$
$$= \int_{B(0,1) \cap (X-S)/\gamma} \gamma^{-1} \langle \nabla \psi(u), (JF)_X^T \theta \rangle + \int_{B(0,1) \cap (X-\partial S)/\gamma} \gamma^{-2} \langle \nabla \psi, dN \rangle$$

where the second-to-last expression is a surface integral which we arrived at by applying the divergence theorem, using that the Laplacian is the divergence of the gradient, and in the last step we used that $\psi$ and all of its derivatives vanish on the boundary of the unit sphere.

Using that $\theta = (\theta_1, 0)$ we have

$$\left| \int_{B(0,1) \cap (X-S)/\gamma} \gamma^{-1} \langle \nabla \psi(u), (JF)_X^T \theta \rangle \right| \le \gamma^{-1} \int_{B(0,1)} \|\nabla \psi(u)\| \|(JF_1)_X\|_{OP} \|\theta\| \tag{16}$$
$$\le 8^d \gamma^{-1} \|(JF_1)_X\|_{OP} \|\theta\|. \tag{17}$$

Let $p$ be the point in $\partial(X-S)/\gamma$ which is closest in Euclidean distance to the origin. Let $n(q)$ denote the unit normal vector at point $q$ oriented outwards (Gauss map). Note that by first-order optimality conditions for $p$, we must have $n(p) = p/\|p\|$. Since $dN = n(q)dA$ where $dA$ is the surface area form, we have

$$\int_{B(0,1) \cap (X-\partial S)/\gamma} \langle \nabla \psi, dN \rangle = \int_{q \in B(0,1) \cap (X-\partial S)/\gamma} \langle \nabla \psi(q), n(p) + (n(q) - n(p)) \rangle dA$$
$$= \int_{q \in B(0,1) \cap (X-\partial S)/\gamma} \frac{-2\psi(q)}{(1 - \|q\|^2)^2} \langle q, \frac{p}{\|p\|} + (n(q) - n(p)) \rangle dA.$$

We now show how to lower bounding the integral by showing $\langle q, \frac{p}{\|p\|} + (n(q) - n(p)) \rangle$ is lower bounded.

Let $c(t)$ be a minimal unit-speed geodesic on $\mathcal{M} := (X-\partial S)/\gamma$ from $p$ to $q$. Note that $\tau_{\mathcal{M}} = \tau_{\partial S}/\gamma$ so if $\gamma$ is very small, $\mathcal{M}$ is very well-conditioned. By the fundamental theorem of calculus, we have that

$$\langle p, q \rangle = \langle p, p \rangle + \int_0^1 \langle p, c'(t) \rangle dt = \langle p, p \rangle + \int_0^1 \langle \text{Proj}_{T_{c(t)}} p, c'(t) \rangle dt$$

where $T_{c(t)}$ is the tangent space to $\mathcal{M}$ at the point $c(t)$. Hence by the Cauchy-Schwarz inequality we have

$$|\langle p, q \rangle \ge \langle p, p \rangle - \int_0^1 \|\text{Proj}_{T_{c(t)}} p\| \|c'(t)\| dt.$$

By Proposition 6.3 of Niyogi et al. (2008), we have that for $\phi_t$ the angle between the tangent spaces $T_p$ and $T_{c(t)}$ that

$$\cos \phi_t \ge 1 - \frac{1}{\tau_{\mathcal{M}}} d_{\mathcal{M}}(p, c(t)) = 1 - \frac{t}{\tau_{\mathcal{M}}} d_{\mathcal{M}}(p, q). \tag{18}$$

Since $\sin^2 \phi_t + \cos^2 \phi_t = 1$ and $p$ is orthogonal to the tangent space at $T_p$, it follows that

$$\| \operatorname{Proj}_{T_{c(t)}} p \| \leq \|p\| |\sin \phi_t| = \|p\| \sqrt{1 - \cos^2 \phi_t} \leq \|p\| \sqrt{(2t/\tau_{\mathcal{M}}) d_{\mathcal{M}}(p,q) + (t/\tau_{\mathcal{M}})^2 d_{\mathcal{M}}(p,q)^2}$$
$$\leq \|p\| \sqrt{(2t/\tau_{\mathcal{M}}) d_{\mathcal{M}}(p,q)} + \|p\| (t/\tau_{\mathcal{M}}) d_{\mathcal{M}}(p,q)$$

hence

$$\int_0^1 \| \operatorname{Proj}_{T_{c(t)}} p \| \|c'(t)\| dt \leq (2/3) \|p\| \sqrt{(2/\tau_{\mathcal{M}})} d_{\mathcal{M}}(p,q)^{3/2} + \|p\| (1/2\tau_{\mathcal{M}}) d_{\mathcal{M}}(p,q)^2.$$

Since $\|p - q\| \leq 2$, provided that $\tau_{\mathcal{M}} > 16$ we have by Proposition 6.3 of Niyogi et al. (2008) that

$$d_{\mathcal{M}}(p,q) \leq \tau_{\mathcal{M}} (1 - \sqrt{1 - 2\|p - q\|/\tau_{\mathcal{M}}}) \leq 4.$$

Combining, we have for some absolute constant $C > 0$ that

$$\langle p, q \rangle \geq \langle p, p \rangle (1 - C\sqrt{1/\tau_{\mathcal{M}}} - C/\tau_{\mathcal{M}}).$$

Also, we can compute

$$\|n(q) - n(p)\| = \sqrt{2 - 2\cos \phi_1} \leq \sqrt{\frac{2}{\tau_{\mathcal{M}}} d_{\mathcal{M}}(p,q)} \leq \sqrt{\frac{8}{\tau_{\mathcal{M}}}}$$

so

$$|\langle q, n(q) - n(p) \rangle\rangle| \leq \|q\| \|n(q) - n(p)\| \leq \sqrt{\frac{8}{\tau_{\mathcal{M}}}}.$$

Hence provided $\tau_{\mathcal{M}} > C'$ for some absolute constant $C' > 0$ and $\|p\| > 0.1$, we have

$$\left| \int_{q \in B(0,1) \cap (X - \partial S)/\gamma} \frac{-2\psi(q)}{(1 - \|q\|^2)^2} \langle q, \frac{p}{\|p\|} + (n(q) - n(p)) \rangle dA \right|$$
$$\geq \int_{q \in B(0,1) \cap (X - \partial S)/\gamma} \frac{\psi(q)}{(1 - \|q\|^2)^2} \|p\| dA$$

using that the integrand on the left is always negative. We can further lower bound the integral by considering the intersection of $\mathcal{M}$ with a ball of radius $r := \frac{1 - \|p\|}{2}$ centered at $p$. We have

$$\int_{q \in B(0,1) \cap (X - \partial S)/\gamma} \frac{\psi(q)}{(1 - \|q\|^2)^2} \|p\| dA \geq \int_{q \in B(p,r) \cap \mathcal{M}} \frac{\psi(q)}{(1 - \|q\|^2)^2} \|p\| dA$$
$$\geq \|p\| (\cos \theta)^k vol(B^k(p,r)) \inf_{q \in B(p,r) \cap \mathcal{M}} \frac{\psi(q)}{(1 - \|q\|^2)^2}$$
$$= \|p\| (\cos \theta)^k r^k \inf_{q \in B(p,r) \cap \mathcal{M}} \frac{B_k \psi(q)}{(1 - \|q\|^2)^2}$$

where $k = d - 1$ is the dimension of $\mathcal{M}$ and $\theta = \arcsin(r/2\tau)$ and we applied Lemma 5.3 of Niyogi et al. (2008). If $\|p\| \in (0.1, 0.9)$ this is lower bounded by a constant $C_k > 0$ which is at worst exponentially small in $k$.

Hence recalling (17) we have for any $X$ with $d(X, \partial S) \in (0.1\gamma, 0.9\gamma)$ and for $\gamma$ sufficiently small so that $\gamma 8^{k+1} \|(JF_1)_X\|_{OP} \|\theta\| < C_k/4$ for any such $X$, we have that

$$\left( (\nabla F_2)_X^T (JF)_X^T \theta + \Delta F_2 \right)^2 \geq \gamma^{-4} C_k'$$

where $C_k' > 0$ is a constant that is at worst exponentially small in $k$. Therefore

$$\mathbb{E}_{X | d(X, \partial S) \in [-\gamma, \gamma]} \left( (\nabla F_2)_X^T (JF)_X^T \theta + \Delta F_2 \right)^2 \geq \gamma^{-4} C_k' \frac{\Pr(d(X, \partial S) \in (0.1\gamma, 0.9\gamma))}{\Pr(d(X, \partial S) \leq \gamma)}.$$

Combining these estimates, we have for some constant $C_k'' > 0$ which is at worst exponentially small in $k$ and $\gamma$ sufficiently small (to satisfy the conditions above, including the requirement $\tau_{\mathcal{M}} > C''$) that

$$\lambda_{max}(\Gamma_{\text{SM}}) \geq \frac{C_k'' \Pr(d(X, \partial S) \in (0.1\gamma, 0.9\gamma))}{\Pr(d(X, \partial S) \leq \gamma)^2}. \tag{19}$$

Observe that for any points $x, y$ and $\theta = (\theta_1, 0)$ we have by the mean value theorem that

$$p_\theta(x)/p_\theta(y) = \exp\left(\langle\theta_1, F_1(x) - F_1(y)\rangle\right) \leq \exp\left(\|\theta\|\sup_{\theta\in[0,1]}\|(JF_1)_{\theta x+(1-\theta)y}\|_{OP}\|x-y\|\right).$$
(20)

so the log of the density is Lipschitz. This basically reduces estimating $\Pr(d(X, \partial S) \leq \gamma)$ for small $\gamma$ to understanding the volume of tubes around $\partial S$, which can be done using the same ideas as the proof of Weyl's tube formula (Weyl, 1939; Gray, 2003).

**Lemma 6** (Proposition 6.1 of Niyogi et al. (2008)). *Let $\mathcal{M}$ be a smooth and compact submanifold of dimension $q$ in $\mathbb{R}^d$. At a point $p \in \mathcal{M}$ let $B : T_p \times T_p \to T_p^\perp$ denote the second fundamental form, and for a unit normal vector $u$, let $L_u$ be the linear operator defined so that $\langle u, B(v, w)\rangle = \langle v, L_u w\rangle$ (this matches the notation from Niyogi et al. (2008)). Then*

$$\|L_u\|_{OP} \leq \frac{1}{\tau_\mathcal{M}}.$$

**Lemma 7** (Lemma 3.14 of Gray (2003)). *Let $\mathcal{M}$ be a smooth and compact submanifold of dimension $q$ in $\mathbb{R}^d$. Let $\exp_p$ denote the exponential map from the normal bundle at $p$. The Jacobian determinant of the map*

$$\mathcal{M} \times (-1/\tau_\mathcal{M}, 1/\tau_\mathcal{M}) \times S_{d-q-1} \to \mathbb{R}^d, \quad (p, t, u) \mapsto \exp_p(tu)$$

*is $\det(I - tL_u)$.*

We can compute

$$\Pr(d(X, \partial S) \leq r) = \int_{x:d(x,\partial S)\leq r} p_\theta(x)dx = \int_{p\in\partial S}\int_0^r\int_{S_0}\det(I - tL_u)p_\theta(\exp_p(tu))\,du\,dt\,dA$$

where in the second equality we performed a change of variables and obtained the result by applying Lemma 7. We have

$$\det(I - tL_u) \in [(1 - t/\tau)^k, (1 + t/\tau)^k]$$

and so applying (20) we find that if we define $c := \gamma\|\theta\|\sup_{x:d(x,\partial S)\leq\gamma}\|(JF_1)_x\|_{OP}$ which can be made arbitrarily small by taking $\gamma$ sufficiently small, then

$$\Pr(d(X, \partial S) \leq r) \in [2e^{-c}\gamma(1 - \gamma/\tau)^kV, 2e^c\gamma(1 + \gamma/\tau)^kV]$$
(21)

where

$$V := \int_{\partial S} p(x)dA.$$

Note that $(1 + \gamma/\tau)^k \leq e^{k\gamma/\tau}$ and $(1 - \gamma/\tau)^k \geq \exp(-O(\gamma k/\tau))$ provided that $\gamma/\tau = O(1/k)$. Since $\Pr(d(X, \partial S) \in (0.1\gamma, 0.9\gamma)) = \Pr(d(X, \partial S) < 0.9\gamma) - \Pr(d(X, \partial S) \leq 0.1\gamma)$ and the distribution we consider has a density, by combining (21) and (19) we find that for $\gamma$ sufficiently small we have

$$\lambda_{max}(\Gamma_{\text{SM}}) \geq C_k'''\frac{1}{\gamma\int_{\partial S}p(x)dA}$$

where $C_k'''$ is at worst exponentially small in $k$. □

## E.2 RELATING FISHER MATRICES OF AUGMENTED AND ORIGINAL SUFFICIENT STATISTICS

Next, we show that adding the extra sufficient statistic $F_2$ has a comparatively minor effect on the efficiency of MLE. Intuitively, to be able to estimate the coefficient of $F_2$ correctly we just need: (1) the variance of $F_2$ is large, so that a nonzero coefficient of $F_2$ can be observed from samples (e.g. when $F_2$ encodes the cut $S$, the coefficient can be estimated by looking at the relative weight between $S$ and $S^C$), and (2) there is no redundancy in the sufficient statistics, e.g. $F_2 \neq F_1$ since otherwise different coefficients can encode the same distribution. The proof of this uses that the inverse covariance of the MLE has a simple explicit form (the Fisher information, which is the covariance matrix of $(F_1, F_2)$), and conditions (1) and (2) naturally appear when we use this fact.

Quantitatively, we show:

**Lemma 8.** *Suppose that $F = (F_1, F_2)$ is a random vector valued in $\mathbb{R}^{m+1}$ with $F_1$ valued in $\mathbb{R}^m$ and $F_2$ valued in $\mathbb{R}$. Suppose that $F_2$ is not in the affine of linear combinations of the coordinates of $F_1$, i.e. for all $w_1 \in \mathbb{R}_m$ there exists $\delta > 0$ such that*

$$\mathrm{Cov}(\langle w_1, F_1 \rangle, F_2)^2 \leq \delta \mathrm{Var}(\langle w_1, F_1 \rangle) \mathrm{Var}(F_2).$$

*Then we have the lower bound*

$$\Sigma_F \succeq (1 - \delta) \begin{bmatrix} \Sigma_{F_1} & 0 \\ 0 & \mathrm{Var}(F_2) \end{bmatrix}$$

*in the standard PSD (positive semidefinite) order.*

*Proof.* To show a lower bound on

$$\Sigma_F = \begin{bmatrix} \Sigma_{F_1} & \Sigma_{F_1 F_2} \\ \Sigma_{F_2 F_1} & \Sigma_{F_2} \end{bmatrix}$$

observe that

$$\langle w, \Sigma_F w \rangle = \langle w_1, \Sigma_{F_1} w_1 \rangle + 2 w_2 \langle w_1, \Sigma_{F_1 F_2} \rangle + w_2^2 \Sigma_{F_2}$$

so under the assumption we have by the AM-GM inequality that

$$\langle w, \Sigma_F w \rangle \geq (1 - \delta) [\langle w_1, \Sigma_{F_1} w_1 \rangle + w_2^2 \Sigma_{F_2}]$$

and hence $\Sigma_F$ is lower bounded in the PSD order as long as $\Sigma_{F_1}$ is and $\Sigma_{F_2}$ is. $\qquad\square$

The lower bound on $\mathrm{Var}(F_2)$ is guaranteed when $F_2$ corresponds to a cut with large mass on both sides since the variance of $F_2$ is lower bounded by its variance conditioned on being away from the boundary of $S$.

### E.3 PUTTING TOGETHER

Finally, given Lemma 5 and 8, we can complete the proof of Theorem 3.

*Proof of Theorem 3.* Define $\rho = \mathrm{Pr}(X \in S)$ for the purpose of this proof. Observe that by (21)

$$\mathrm{Var}(1_S - F_2) \leq \mathbb{E}(1_S - F_2)^2 \leq \mathrm{Pr}(d(X, \partial S) \leq \gamma) \leq 4\gamma V$$

where $V = \int_{\partial S} p(x) dA$. We have that

$$\mathrm{Cov}(\langle w_1, F_1 \rangle, F_2) = \mathrm{Cov}(\langle w_1, F_1 \rangle, 1_S) + \mathrm{Cov}(\langle w_1, F_1 \rangle, F_2 - 1_S)$$

so if $w_1$ is arbitrary and normalized so that $\mathrm{Var}(\langle w_1, F_1 \rangle) = 1$ then we have

$$|\mathrm{Cov}(\langle w_1, F_1 \rangle, F_2)| \leq \sqrt{1 - \delta_1} \sqrt{\mathrm{Var}(1_S)} + \sqrt{\mathrm{Var}(F_2 - 1_S)}$$

$$\leq \left( \sqrt{1 - \delta_1} + 2\sqrt{\frac{\gamma V}{\rho(1 - \rho)}} \right) \sqrt{\mathrm{Var}(1_S)}.$$

Therefore provided $\delta > 0$ we have

$$\Sigma_F^{-1} \preceq \frac{1}{\delta} \begin{bmatrix} \Sigma_{F_1}^{-1} & 0 \\ 0 & \mathrm{Var}(F_2)^{-1} \end{bmatrix}.$$

On the other hand, by Lemma 5 we have

$$\lambda_{max}(\Gamma_{\mathrm{SM}}) \geq \frac{c^d}{\gamma V}.$$

Hence there exists some $w$ such that

$$\frac{\sigma_{\mathrm{SM}}^2(w)}{\sigma_{\mathrm{MLE}}^2(w)} \geq \frac{\delta c^d}{\max\{\|\Sigma_{F_1}^{-1}\|_{OP}, 1/\rho(1 - \rho)\}} \frac{1}{\gamma V} \geq \frac{\delta c^d}{1 + \rho(1 - \rho)\|\Sigma_{F_1}^{-1}\|_{OP}} \frac{\rho(1 - \rho)}{\gamma V}.$$

Using that $\min\{\rho, 1 - \rho\}/2 \leq \rho(1 - \rho) \leq 1/4$ and dividing $c$ by two gives the result. $\qquad\square$

### E.4 Multimodal example: Proof of Corollary 1

*Proof of Corollary 1.* First observe that

$$\int_{-\infty}^{\infty} e^{-F_1(x)}dx = 2\int_0^{\infty} e^{-(1/8)(x-a)^2(x/a+1)^2}dx \le 2\int_{-\infty}^{\infty} e^{-(1/8)(x-a)^2}dx$$

$$= 2\int_{-\infty}^{\infty} e^{-(x^2/8)}dx =: C$$

where $C$ is a positive constant independent of $a$. Using that $F_1(x) = (1/8)(x-a)^2(x/a+1)^2$ it then follows that

$$\Pr(X \in [a-1, a+1]) = \frac{\int_{a-1}^{a+1} e^{-F_1(x)}dx}{\int_{-\infty}^{\infty} e^{-F_1(x)}dx} \ge \frac{e^{-(1/8)(x/a+1)^2}}{C} \ge C' > 0$$

where $C'$ is a positive constant independent of $a$. From this, we see by the law of total variance that $\mathrm{Var}(F_1) \ge \mathrm{Var}(F_1 \mid X \in [a-1, a+1])\Pr(X \in [a-1, a+1]) \ge C'' > 0$ where $C'' > 0$ is another positive constant independent of $a$. Hence $\|\Sigma_{F_1}^{-1}\|_{OP} = O(1)$ independent of $a$. Also, if we define $S = \{x : x > 0\}$ then

$$\mathrm{Cov}(F_1(x), 1_S) = 0$$

becuase $F_1(x)$ is even, $1_S$ is odd and the distribution is symmetric about zero. So we can take $\delta_1 = 1$ in the statement of Theorem 3.

Therefore, applying Theorem 3 to $S$ and using that $F_1(0) = -a^2/8$, we therefore get for $\gamma$ smaller than an absolute constant, that the inefficiency is lower bounded by $\Omega(e^{a^2/8}/\gamma)$. By taking $\gamma$ equal to a fixed constant we get the result. □

In Section 7, we perform simulations which show the performance of score matching indeed degrades exponentially as $a$ beomes large.

## F  Discrete Analogues of Score Matching

**Glauber dynamics.**  The Glauber dynamics or Gibbs sampler is the standard sampler for discrete spin systems — it repeatedly selects a random coordinate and then resamples the spin $X_i$ there conditional on all of the other ones (i.e. conditional on $X_{\sim i}$). See e.g. Levin and Peres (2017). This is the standard sampler for discrete systems, but it also applies and has been extensively studied for continuous ones (see e.g. Marton (2013)). Exponential ergodicity of the Glauber dynamics is equivalent to the Modified Log-Sobolev Inequality (MLSI) — in most cases where the MLSI is known, approximate tensorization of entropy is also, e.g. Chen et al. (2021b); Anari et al. (2021a); Marton (2015); Caputo et al. (2015).

### F.1 Finite sample bounds

We state explicitly the analogue of Theorem 1 for pseudolikelihood, which follows from the same proof by replacing Proposition 1 with Proposition 4.

**Theorem 5.** *Suppose that $\mathcal{P}$ is a class of probability distributions containing $p$ and define $C_{AT}(\mathcal{P}, \mathcal{P}) := \sup_{q \in \mathcal{P}} C_{AT}(q, \mathcal{P}) \le \sup_{q \in \mathcal{P}} C_{AT}(q)$ to be the worst-case (restricted) approximate tensorization constant in the class of distributions. Let*

$$\mathcal{R}_n := \mathbb{E}_{X_1, \ldots, X_n, \epsilon_1, \ldots, \epsilon_n} \sup_{q \in \mathcal{P}} \frac{1}{n} \sum_{i=1}^n \epsilon_i \left[ \sum_{j=1}^d \log q((X_i)_j \mid (X_i)_{\sim j}) \right]$$

*be the expected Rademacher complexity of the class given $n$ samples $X_1, \ldots, X_n \sim p$ i.i.d. and independent $\epsilon_1, \ldots, \epsilon_n \sim Uni\{\pm 1\}$ i.i.d. Rademacher random variables. Let $\hat{p}$ be the pseudolikelihood estimator from $n$ samples, i.e. $\hat{p} = \arg\min_{q \in \mathcal{P}} \hat{L}_p(q)$. Then*

$$\mathbb{E}\,\mathbf{KL}(p, \hat{p}) \le 2C_{AT}(\mathcal{P}, \mathcal{P})\mathcal{R}_n.$$

*In particular, if $C_{AT} < \infty$ then $\lim_{n \to \infty} \mathbb{E}\,\mathbf{KL}(p, \hat{p}) = 0$ as long as $\lim_{n \to \infty} \mathcal{R}_n = 0$.*

## F.2 RATIO MATCHING AND APPROXIMATE TENSORIZATION

Marton (2015) studied a strengthened version of approximate tensorization of the form

$$\mathbf{KL}(p,q) \leq C_{AT2}(q) \sum_{i=1}^{d} \mathbb{E}_{X_{\sim i} \sim p_{\sim i}} \mathbf{TV}^2(p(X_i \mid X_{\sim i}), q(X_i \mid X_{\sim i})) \tag{22}$$

where $\mathbf{TV}$ denotes the total variation distance (see Cover (1999)). (This is known to hold for a class of distributions $q$ satisfying a version of Dobrushin's condition and marginal bounds (Marton, 2015).) This inequality is stronger than the standard approximate tensorization because of Pinsker's inequality $\mathbf{TV}^2(P,Q) \lesssim \mathbf{KL}(P,Q)$ (Cover, 1999). In the case of distributions on the hypercube, we have

$$\begin{aligned}
\mathbf{TV}^2&(p(X_i \mid X_{\sim i}), q(X_i \mid X_{\sim i})) \\
&= |p(X_i = +1 \mid X_{\sim i}) - q(X_i = +1 \mid X_{\sim i})|^2 \\
&= \mathbb{E}_{X_i \sim p_{X_i \mid X_{\sim i}}} |1(X_i = +1) - q(X_i = +1 \mid X_{\sim i})|^2 \\
&\quad - \mathbb{E}_{X_i \sim p_{X_i \mid X_{\sim i}}} |1(X_i = +1) - p(X_i = +1 \mid X_{\sim i})|^2
\end{aligned}$$

where in the last step we used the Pythagorean theorem applied to the $p_{X_i \mid X_{\sim i}}$-orthogonal decomposition

$$\begin{aligned}
1(X_i = +1) - q(X_i = +1 \mid X_{\sim i}) &= [1(X_i = +1) - p(X_i = +1 \mid X_{\sim i})] \\
&\quad + [p(X_i = +1 \mid X_{\sim i}) - q(X_i = +1 \mid X_{\sim i})]
\end{aligned}$$

Hence, there exists a constant $K_p'$ not depending on $q$ such that

$$\sum_{i=1}^{d} \mathbb{E}_{X_{\sim i} \sim p_{\sim i}} \mathbf{TV}^2(p(X_i \mid X_{\sim i}), q(X_i \mid X_{\sim i})) = K_p + M_p(q) \tag{23}$$

where we define the ratio matching objective function to be

$$M_p(q) := \sum_{i=1}^{d} \mathbb{E}_{X \sim p} |1(X_i = +1) - q(X_i = +1 \mid X_{\sim i})|^2 \tag{24}$$

This objective is now straightforward to estimate from data, by replacing the expectation with the average over data. Analogous to before, we have the following proposition:

**Proposition 5.** *We have*
$$\mathbf{KL}(p,q) \leq C_{AT2}(q)(M_p(q) - M_p(p))$$
*and more generally for any class $\mathcal{P}$ containing $p$, we have $\mathbf{KL}(p,q) \leq C_{AT2}(q, \mathcal{P})(M_p(q) - M_p(p))$.*

We now show how to rewrite $M_p(q)$ to match the formula from the original reference. Observe

$$M_p(q) = \frac{1}{4} \sum_{i=1}^{d} \mathbb{E}_{X \sim p} |X_i - \mathbb{E}_q[X_i \mid X_{\sim i}]|^2 = \frac{1}{4} \sum_{i=1}^{d} \mathbb{E}_{X \sim p} |1 - X_i \mathbb{E}_q[X_i \mid X_{\sim i}]|^2$$

Observe that for any $z \in \{\pm 1\}$ we have

$$z\mathbb{E}_q[X_i \mid X_{\sim i}] = \frac{q(X_i = z \mid X_{\sim i}) - q(X_i = -z \mid X_{\sim i})}{q(X_i = z \mid X_{\sim i}) + q(X_i = -z \mid X_{\sim i})}$$

and

$$\begin{aligned}
1 - z\mathbb{E}_q[X_i \mid X_{\sim i}] &= \frac{2q(X_i = -z \mid X_{\sim i})}{q(X_i = z \mid X_{\sim i}) + q(X_i = -z \mid X_{\sim i})} \\
&= \frac{2}{1 + q(X_i = z \mid X_{\sim i})/q(X_i = -z \mid X_{\sim i})}.
\end{aligned}$$

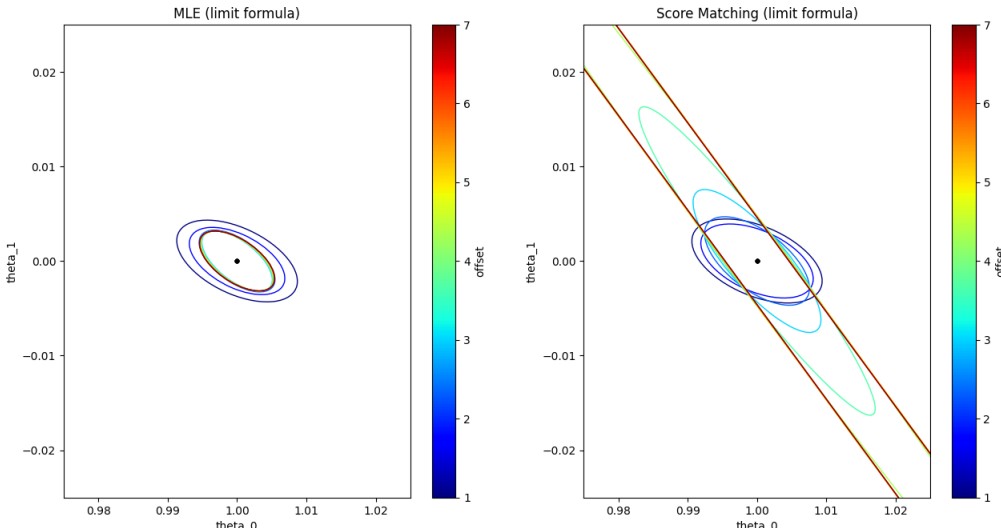

Figure 3: Level sets for the distribution over estimates in the same example as Figure 1. We see that as the distance $a$ between modes increases, the direction of large variance for the score matching estimator (right figure) corresponds to the difference of the sufficient statistics which encodes the sparse cut in the distribution. On the other hand, the MLE (left figure) does not exhibit this behavior and has low variance in all directions.

Also for $z \in \{\pm 1\}^d$ we have $q(X_i = z_i \mid X_{\sim i} = z_{\sim i})/q(X_i = -z_i \mid X_{\sim i} = z_{\sim i}) = q(z)/q(z_{-i})$ where $z_{-i}$ reprsents $z$ with coordinate $i$ flipped, so

$$M_p(q) = \sum_{i=1}^{d} \mathbb{E}_{X \sim p} \left( \frac{1}{1 + q(X)/q(X_{-i})} \right)^2$$

which matches the formula in Theorem 1 of Hyvärinen (2007).

Summarizing, minimizing the ratio matching objective makes the right hand side of the strengthened tensorization estimate (22) small, so when $C_{AT2}(q)$ is small it will imply successful distribution learing in KL. (The obvious variant of Theorem 5 will therefore hold.) In this way ratio matching can also be understood as a relaxation of maximum likelihood.

## G    FURTHER SIMULATIONS

**Complementary visualization to Figure 1.**    In Figure 3, we illustrate the distribution of the errors in the bimodal experiment with the cut statistic. As expected based on the theory, the direction where score matching with large offset performs very poorly corresponds to the difference between the two sufficient statistics, which encodes the sparse cut in the distribution.

**Fitting a bimodal distribution without a cut statistic.**    In Figure 4 we show the result of fitting the same bimodal distribution using score matching, but we remove the second sufficient statistic (which is correlated with the sparse cut in the distribution). In this case, score matching fits the distribution nearly as well as the MLE. This is consistent with our theory (e.g. the failure of score matching in Theorem 3 requires that we have a sufficient statistic approximately representing the cut) and justifies some of the distinctions we made in our results: even though the Poincaré constant is very large, the asymptotic variance of score matching within the exponential family is upper bounded by the *restricted* Poincaré constant (see Theorem 2) which is much smaller.

**Example 3** (Application of Theorem 2 to this example). *To briefly expand the last point, we show how to apply Theorem 2 in this example (Example 2, where we have **not** added a bad cut statistic.) The restricted Poincaré constant for applying Theorem 2 will be*

$$C := \frac{\mathrm{Var}(F_1(X))}{\mathbb{E}(F_1'(X))^2} = \frac{\mathrm{Var}(X^2 - X^4/2a^2)}{\mathbb{E}(2X - 2X^3/a^2)^2} \tag{25}$$

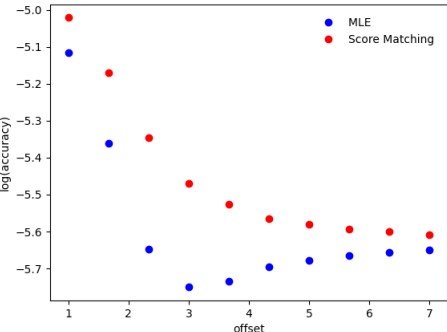

Figure 4: Here we see the result of running an identical experiment to Figure 1, only we remove the second sufficient statistic, so our distribution is now $p_\theta(x) \propto e^{\theta_0(x^2 - x^4/(2a^2))}$ where $\theta_0 = 1$ and we again vary the offset $a$ between 1 and 7. With only the single sufficient statistic, score matching performs comparably to MLE.

*which asymptotically goes to a constant, rather than blowing up exponentially, as $a$ goes to infinity. (This can be made formal using arguments as in the proof of Corollary 1; informally, the distribution is similar to a mixture of two standard Gaussians centered at $\pm a$ so the numerator is close to $\mathrm{Var}_{Z \sim N(0,1)}((a+Z)^2 - (a+Z)^4/2a^2) = \mathrm{Var}(2aZ + Z^2 - (4aZ + 6Z^2 + 4Z^3/a + Z^4)/2) = \Theta(1)$ and the denominator is approximately $\mathbb{E}_{Z \sim N(0,1)}(2(a+Z) - 2(a+Z)^3/a^2)^2 = \mathbb{E}(2Z - 2(3Z + 3Z^3/a + Z^3/a^2))^2 = \Theta(1).)*

*Given this bound on the restricted Poincaré constant, we can apply Theorem 2. Based on similar reasoning to above, one can show that $\mathbb{E}F_1'(X)^4 = (-1/4a^2)^4 \mathbb{E}((X-a)(X+a)^2 + (X-a)^2(X+a))^4 = \Theta(1)$ and $\mathbb{E}F_1''(X)^2 = \mathbb{E}(-3x^2/2a^2 + 1/2)^2 = \Theta(1)$, so we conclude that $\|\Gamma_{SM}\|_{OP} = O(\|\Gamma_{MLE}\|_{OP}^2)$. This proves that score matching will perform not much worse than the MLE, as we saw in the experimental result of Figure 4.*

**Remark 8.** *Example 3 shows a case where there is a large gap between the restricted and unrestricted Poincaré constants. This also implies a completely analogous gap between appropriate restricted and unrestricted log-Sobolev constants, as used e.g. in the context of Theorem 1. To elaborate, we know that the unrestricted log-Sobolev constant blows up exponentially in $a$, just like the unrestricted Poincaré constant, because $C_{LS} \geq C_P/2$ (Van Handel, 2014). On the other hand, if we fix the ground truth distribution $p_a$ consider the class of distributions*

$$\mathcal{P}_r = \{p_{a'} : |a - a'| \leq r\},$$

*we have that*

$$\lim_{r \to 0} C_{LS}(q, \mathcal{P}_r) = C/2$$

*where $C$ is the constant defined in (25) in terms of $a$ (and which is $O(1)$ as $a \to \infty$). This is because from the definition as an exponential family, we have*

$$p_a(x)/p_{a'}(x) = \frac{\exp\left((a - a')F_1(x)\right)}{\mathbb{E}_{a'} \exp\left((a - a')F_1(x)\right)}$$

*so*

$$\lim_{a' \to a} \frac{\mathbf{KL}(p_a, p_{a'})}{\mathcal{I}(p_a \mid p_{a'})} = \lim_{a' \to a} \frac{(a - a')^2 \mathrm{Var}_{p_{a'}}(F_1(x))}{2(a - a')^2 \mathbb{E}_{p_{a'}} \|\nabla F_1(x)\|^2} = C/2$$

*where the first equality is by a standard Taylor expansion argument (see proof of Lemma 3.28 of (Van Handel, 2014)).*

**Fitting a unimodal but not smooth distribution.** In Figure 5, we demonstrate what happens when the distribution is unimodal (and has small isoperimetric constant), but the sufficient statistic is not quantitatively smooth. More precisely, we consider the case $p_\theta(x) \propto e^{-\theta_0 x^2/2 - \theta_1 \sin(\omega x)}$ as $\omega$ increases. In the figure, we used the formulas from asymptotic normality to calculate the distribution

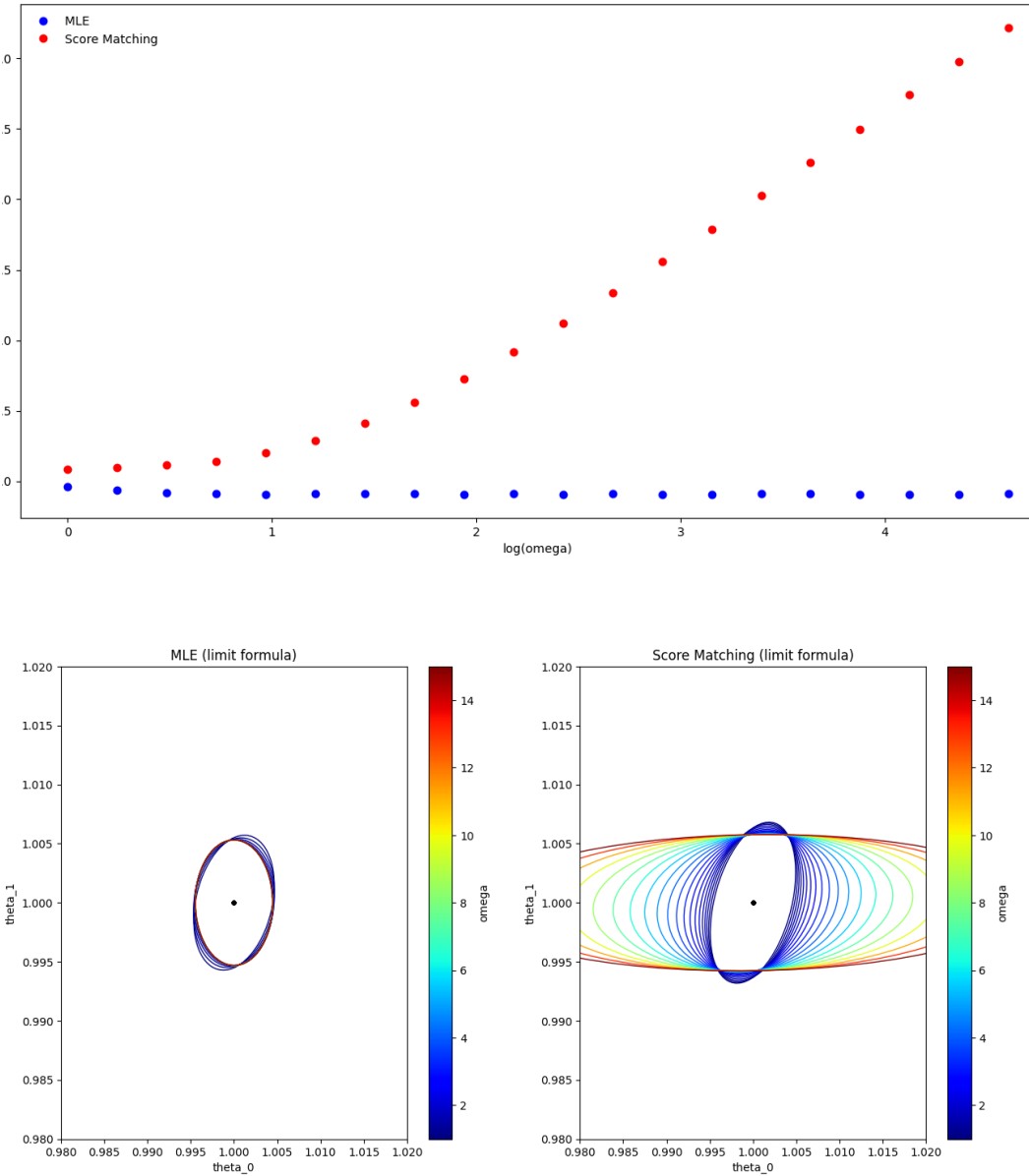

Figure 5: Score matching vs MLE for a distribution with a rapidly oscillating sufficient statistic, $p_\theta(x) \propto e^{-\theta_0 x^2/2 - \theta_1 \sin(\omega x)}$ where $(\theta_0, \theta_1) = (1, 1)$, and increasing $\omega$. On the top, for increasing $\omega$ we show a log-log plot of the average Euclidean distance in parameter space between $\theta$ and the output of each estimator. On the bottom, for each value of $\omega$, we draw a level set of the distribution within which a fixed fraction of returned estimates lie (MLE left, score matching right). Score matching becomes increasingly inaccurate as $\omega$ increases while the MLE stays extremely accurate.

over parameter estimates from 100,000 samples. We also verified via simulations that the asymptotic formula almost exactly matches the actual error distribution.

The result is that while the MLE can always estimate the coefficient $\theta_1$ accurately, score matching performs much worse for large values of $\omega$. This demonstrates that the dependence on smoothness in our results (in particular, Theorem 2) is actually required, rather than being an artifact of the proof. Conceptually, the reason score matching fails even when though the distribution has no sparse cuts is this: the gradient of the log density becomes harder to fit as the distribution becomes less smooth (for example, the Rademacher complexity from Theorem 1 will become larger as it scales with $\nabla_x \log p$ and $\nabla_x^2 \log p$).

**Fitting a mixture of Gaussians with a one-layer network: further discussion.** We provide some further remarks on the results in Figure 2. In the right hand side example (the one with large separation between modes), the shape of the two Gaussian components is learned essentially perfectly — it is only the relative weights of the two components which are wrong. This closely matches the idea behind the proof of the lower bound in Theorem 3; informally, the feedforward network can naturally represent a function which detects the cut between the two modes of the distribution, i.e. the additional bad sufficient statistic $F_2$ from Theorem 3. The fact that the shapes are almost perfectly fit where the distribution is concentrated indicates that the *test loss $J_p$* is near its minimum. Recall from (1) that the suboptimality of a distribution $q$ in score matching loss is given by $J_p(q) - J_p(p) = \mathbb{E}_p \|\nabla \log p - \nabla \log q\|^2$. If we let $q$ be the distribution recovered by score matching, we see from the figure that the slopes of the distribution were correctly fit wherever $p$ is concentrated, so $\mathbb{E}_p \|\nabla \log p - \nabla \log q\|^2$ is small. However near-optimality of the test loss $J_p(q)$ does not imply that $q$ is actually close to $p$: the test loss does not heavily depend on the behavior of $\log q$ in between the two modes, but the value of $\nabla \log q$ in between the modes affects the relative weight of the two modes of the distribution, leading to failure.

*Model details:* both models illustrated in the figure have 2048 $\mathtt{tanh}$ units and are trained via SGD on fresh samples for 300000 steps. After training the model, the estimated distribution is computed from the learned score function using numerical integration.

