# OpenReview forum: "Statistical Efficiency of Score Matching: The View from Isoperimetry"
_ICLR.cc/2023/Conference — ICLR 2023 notable top 5%_

### Official Review · Reviewer_R2sB · 2022-10-22

**Confidence:** 3
**Correctness:** 4
**Technical Novelty And Significance:** 3
**Empirical Novelty And Significance:** 3
**Recommendation:** 8

**Clarity, Quality, Novelty And Reproducibility:**

Generally, the paper is clear and results are of high quality. Also, the statistical efficiency is a new direction in score estimation, so the paper has novel ideas. Perhaps reproducibility could be improved. I have some comments below that might help improve the paper:
I believe there are a couple of things that need to be clarified. First and foremost, while the paper nicely motivates citing the example how new score estimation techniques circumvent many problems associated with probability density estimation and estimation of partition function, their result does not exactly discuss the property of the score function, say s, but rather the estimated distribution q, which is quite different from the score s, i.e. the derivative of log of q. It is unclear at the moment whether these results translate easily to the properties of score s. That is, as is customary in recent score based generative models, score function, s, is modeled as a neural network and minimizes the loss J as defined in eq. 1 of the paper.  Another important and related matter that needs to be clarified in the paper is that the estimation that the paper analyzes is different from the recent developments in score based generative models where to facilitate the estimation of score, distribution is first convolved with a series of Gaussian noise and score is estimated for all the noisy probability distribution indexed by time. While this paper does not discuss this setup, it is better to explicitly clarify this point since score based generative models are quite common and can immediately confuse readers.

A couple of suggestion for the authors to improve the clarity and presentation:
1. The distribution q first mentioned in section 2, what is it and how is it related to the score?? After reading a couple of sentence, I guessed that the score being estimated is the grad of log of q. But, it is better to clarify that relation and introduce q first. Also, this is very critical here. I am still puzzled why analyzing property of q, other than perhaps that is well studied, is relevant in contrast to studying the property of score directly??
2. From the definition of log Sobolev constant in eq. 2, it seems like the log Sobolev constant does depend on the distribution p. But, later in remark 2, authors state that log Sobolev constant does not depend on p, which is what I was expecting. But, it's better to clarify that in Defn 2 and explain why.

3. The paper is packed with a lot of information, which is hard to unpack for the reader. Perhaps it is better to let go some of the contents and better explain other more relevant ones. For example, how does the score matching estimator in Proposition 2, page 5 come from. Authors have cited the paper by Hyvarinen 2007, but it's better to derive it for clarity.
4. Related to the above comment, pseudolikelihood in section 5 might be a bit less relevant to the analysis of score matching.
5. I found the experiments illuminating and would have liked more of it. Plus, at the moment it only supports theorem 2, something like that for Theorem 1 would have been nice.
6. Lastly,  score estimation in generative models have been linked to log likelihood, which makes me wonder if there is a connection between these results and those. See the following paper:

Chen, T., Liu, G.H. and Theodorou, E.A.. Likelihood Training of Schr\" odinger Bridge using Forward-Backward SDEs Theory. ICLR 2022.


**Strength And Weaknesses:**

Many results of the paper are interesting and insightful. And they are fairly novel since this work is probably the first to discuss the statistical efficiency of score estimator.

However, the clarity of the paper could be improved. I have a few suggestions below. Also, with some more insightful experiments, the motivation of the paper could be boosted.

**Summary Of The Paper:**

The paper analyzes the statistical efficiency of the score estimate obtained by the score matching objective and compares it with the maximum likelihood based approach. Specifically, relation between the loss optimized in score matching J has been compared to the KL divergence between the true distribution p and the estimated distribution q. The paper shows that the bound on the KL depends on the log-Sobolev constant, the score matching estimator converges to a distribution that depends on other quantities like Poincare and smoothness of sufficient statistics. Later this experiments support the main idea of this theorem.

**Summary Of The Review:**

I generally liked the new perspective of analyzing the statistical efficiency of the score matching objective discussed in the paper. Many theoretical results have been presented. However, there were some concerns in presentation and a few clarifying questions. Hence, I am on the positive side.

---

> ### Author Response · Authors · 2022-11-08
> **Thank you for the review! Some clarifications**
>
> Thank you for the positive review and the writing suggestions !  There seem to be a couple of points of confusion, which we hope to clarify.
>
> **“...puzzled why analyzing property of q, other than perhaps that is well studied, is relevant in contrast to studying the property of score directly??”**
>
> This seems to be the main question/point of confusion in your review. Ultimately, the main motivation of score matching is to learn a probability distribution $p$ by fitting the score function $\nabla \log p$ instead of the log-probability, thereby avoiding having to calculate a partition function. However, the final object of interest is still the *learned distribution*—e.g. the score function would be used in conjunction with a Langevin sampler to draw samples from the *learned distribution*. Furthermore, our main motivation is to compare how much statistically less efficient score matching is compared to the (asymptotically optimal) way of learning $p$ via maximum likelihood—which is in turn equivalent to minimizing the KL divergence between the data distribution and the learned distribution. Thus it stands to reason properties of the *distribution* we are learning would be relevant, rather than just the score itself.
>
> **“From the definition of log Sobolev constant in eq. 2, it seems like the log Sobolev constant does depend on the distribution p. But, later in remark 2, authors state that log Sobolev constant does not depend on p, which is what I was expecting. But, it's better to clarify that in Defn 2 and explain why.”**
>
> The Log-Sobolev constant of a distribution $q$ is only a function of the distribution $q$. For example, Definition 2 defines the Log-Sobolev constant of a distribution $q$: and is the smallest constant $C_{LS}(q)$, s.t. *for every other distribution $p$*, eq 2 is satisfied. (That is to say, $C_{LS}(q)$ is only a function of $q$.) The point in Remark 2 is that when the “multiplicative gap” between the score matching objective and the KL divergence objective (eq. 5), when estimating a distribution $p$, depends on the Poincaré constants of the distributions $q$ in the class we are fitting, rather than $p$.
>
> **Clarify we are not talking about denoising score matching:** great suggestion! We added a footnote in the updated draft (in blue, so the difference is easy to see)

---

> > ### Comment · Reviewer_R2sB · 2022-11-21
> > **Thanks for the reply**
> >
> > Thanks for clarifying my confusion. I have updated the score.

---

### Official Review · Reviewer_tSNd · 2022-10-24

**Confidence:** 2
**Correctness:** 3
**Technical Novelty And Significance:** 3
**Empirical Novelty And Significance:** Not applicable
**Recommendation:** 8

**Clarity, Quality, Novelty And Reproducibility:**

The paper is clearly written and easy to follow. The claims seem to be technically solid and novel. I only have a few concerns which are stated above.

**Strength And Weaknesses:**

**Strength:**
1. Score matching is a widely used technique and its statistical analysis is of high interest to the community.
1. Their theoretical results seem novel and technically solid to me. I found the isoperimetric viewpoint quite interesting.
1. I found the paper well-written and easy to follow.

**Major comments:**
1. Why is Thm. 3 about an unknown direction? It makes sense to me if the goal is to estimate $||\theta||$ or $w_0^\top \theta$ for a prescribed $w_0$. The fact that the lower bound is proved for an unknown vector $w$ is confusing to me. Is it possible that there exists some $w'$ such that the score-matching estimator achieves better efficiency?
1. The paper could benefit from adding more explanations and discussions. For example,
    1. How large is $C_{LS}(q)$ compared to $C_{LS}(q, \mathcal{P})$ depending on $\mathcal{P}$? Can you give an example?
    1. Can you provide an example for Thm. 2, just like Thms. 1 and 3?
    1. It would be helpful to give some intuition on the tools used in Sec. 2.
1. From (1), $J_p(q) - J_p(p)$ is equal to $1/2$ of the RHS of (6). Prop. 1 seems to be missing a factor of 2. This also holds true in the latter results like Thm. 1.

**Minor comments:**
1. Should $p$ be smooth in Def. 2?
1. What is $f$ in Def. 3?
1. The notation $I_d$ in the paragraph Mollifiers can be confused with the identity matrix.
1. $J_p(q)$ should be $J_p(p)$ in the equation below (6).
1. The subscripts MLE and SM in Def. 6 and Prop. 2 are too large. Maybe use \text{}?
1. In conclusion, "In tihs paper" --> "In this paper".


**Summary Of The Paper:**

This paper studies the statistical properties of the score matching from a geometry viewpoint. The authors show an upper bound for the KL divergence between the underlying data distribution and the score-matching estimator using the log-Sobolev inequality. Moreover, with a focus on the exponential family, they analyzed the asymptotic efficiency of the score-matching estimator compared to the one of the MLE using The Poincaré inequality and isoperimetric inequality.

**Summary Of The Review:**

This paper provides useful insights into the popular score-matching estimator. The results seem novel and technically solid to me. I thus vote for acceptance.

---

> ### Author Response · Authors · 2022-11-12
> **Thank you for the review!**
>
> Thanks for your careful reading of the paper and positive review! Thank you also for the writing suggestions—we implemented them in the updated draft (changes in blue), and provide clarifications regarding your questions below.
>
> Major comments:
> * **Comment 1: “Why is Thm. 3 about an unknown direction?”**: The high-level goal of Theorem 3 is to show that score matching fails to estimate $\theta$ well. Because $\theta$ is vector-valued, we have a few options to formalize this—for example we could lower bound the squared loss $||\theta - \hat \theta||^2$. Since we show there exists at least one direction where score matching does very poorly, this implies a lower bound on the error in squared norm. What we show is actually a bit more precise than just lower bounding the squared error: our proof identifies a particular direction in which score matching does very poorly. \
> **Comment 1: "Is it possible that there exists some $w’$ such that the score-matching estimator achieves better efficiency?”**: If you mean “better than MLE”—the answer is no, because MLE is asymptotically optimal among all estimators satisfying some regularity conditions (see e.g. Theorem 8.8 in Van der Vaart, Asymptotic statistics, vol 3 (2000)). If you mean to ask if score matching is “less worse” than MLE in some directions, and “more worse” in others—possibly yes, our proof identifies a particularly bad one, which is all we need (see the paragraph above for why this is the case).
>
> * **Comment 2.1**: Yes — we have added a discussion and an example of the gap between $C_{LS}(q)$ and $C_{LS}(q,P)$ in the Remark after the (newly added) Example 3 of Appendix G.
> * **Comment 2.2**: Yes, certainly! We added an example illustrating Theorem 2 following the example corresponding to Figure 4 —this is the (newly added) Example 3 in Appendix G.
> * **Comment 2.3**: We added a sentence on the relationship between log-Sobolev inequality and log-concave distributions. We also have some more exposition and references for the background material in Appendix A. Unfortunately, due to space constraints, it’s difficult to add more in the main part. (If there is something specific you think would be helpful to add let us know. Functional inequalities, as well as their relationship to mixing times of Markov Chains like Langevin dynamics is an area of mathematics with rich history, so it’s difficult to fully do it justice in a couple of pages.)
> * **Comment 3**: Good catch, we fixed the factors of two in the revision.
>
> Minor comments:
> * **Comment 1**: Yes, this was a typo and $p$ should be smooth too—fixed.
> * **Comment 2**: Good catch, it should have said that this inequality needs to hold for all smooth functions $f$. We fixed this in the revision.
> * **Comment 3**: Good point, we renamed it to $G_d$.
> * **Comment 4**: Noted, we updated this as well.
> * **Comment 5**: Thank you, fixed!
> * **Comment 6**: Thank you, fixed!

---

> > ### Comment · Reviewer_tSNd · 2022-11-26
> > **Thanks for the response**
> >
> > Thank the author(s) for their response. I think this is a good paper and keep my score unchanged.

---

### Official Review · Reviewer_gNn2 · 2022-10-25

**Confidence:** 4
**Correctness:** 4
**Technical Novelty And Significance:** 3
**Empirical Novelty And Significance:** 3
**Recommendation:** 8

**Clarity, Quality, Novelty And Reproducibility:**

The paper is well-written, and the proofs are straightforward and easy to follow, even for non-experts in isoperimetric analysis.

To my knowledge, the theoretical connections are novel, though the practical observations they describe have been well-known at the outset of the development of score-based generative models [1].

**Strength And Weaknesses:**

Strengths:
- Proofs are clean and straightforward.
- Theory corroborates with practical observation, that score matching on multimodal data requires noise perturbations.

Weaknesses:
- Theory is mainly descriptive rather than prescriptive --- it is well known that noise-perturbation is crucial for properly learning scores [1].
- Non-asymptotic theory is limited to comparing loss values (KL divergence vs expected square error of scores), which does not provide a clear insight into how the resulting densities themselves compare.
- Asymptotic theory is limited to learning on distributions in the exponential family.

[1] Song, Y. and Ermon, S., 2019. Generative modeling by estimating gradients of the data distribution. Advances in Neural Information Processing Systems, 32.

**Summary Of The Paper:**

The authors provide a theoretical comparison between maximum likelihood estimators and score matching estimators in the case where the distributions are from the exponential family.


**Summary Of The Review:**

This work provides a suitable theoretical premise for the well-known observation in score-based generative modeling that noise-perturbations are critical for good performance. For this reason, the most relevant result to me is Theorem 2, which provides theory for this phenomenon under the (albeit simplistic) assumption that the data distribution is exponential.

---

> ### Author Response · Authors · 2022-11-08
> **Thank you for the positive review!**
>
> Thank you for the positive review and the encouraging words ! We agree that some interesting and exciting future directions would be to provide non-asymptotic results on the comparison between MLE and score matching, as well as provide results beyond an exponential-family parametrization.

---

### Decision · Program_Chairs · 2023-01-20

**Decision:**

Accept: notable-top-5%

**Justification For Why Not Higher Score:**

N/A

**Justification For Why Not Lower Score:**

The paper provides a significant contribution to the understanding of the statistical efficiency of score matching which is of fundamental interest given the widespread use of score matching.

**Metareview: Summary, Strengths And Weaknesses:**

The paper studies the statistical efficiency of the score estimator which was still an open question.
The authors relate such efficiency to properties of the target distribution estimated (such as isoperimetry, and Poincaré inequalities).
Given the high interest in using score matching for learning generative models this work provides significant insight on the properties of score matching estimators and when they fail to be statistically efficient.

All reviewers as well as the AC agree that the paper provides an outstanding theoretical contribution.



**Note From Pc:**

if the above contains the word "oral" or "spotlight" please see: "oral" presentation means -> notable-top-5% and "spotlight" means -> notable-top-25%. As stated in our emails, we are disassociating presentation type from AC recommendations